# Molecular self-assembly strategy tuning a dry crosslinking protein patch for biocompatible and biodegradable haemostatic sealing

Lisha Yu [1,2,8], Zhaodi Liu [1,2,8], Yong Zheng [3], Zongrui Tong[1,2], Yihang Ding [3], Weilin Wang [1,2,4,5,6] ✉, Yuan Ding [1,2,4,5,6] ✉ & Zhengwei Mao [1,2,3,7] ✉

Uncontrolled haemorrhage is a leading cause of trauma-related fatalities, highlighting the critical need for rapid and effective haemostasis. Current haemostatic materials encounter limitations such as slow clotting and weak mechanical strength, while most of bioadhesives compromise their adhesion performance to wet tissues for biocompatibility and degradability. In this study, a molecular self-assembly strategy is proposed, developing a biocompatible and biodegradable protein-based patch with excellent adhesion performance. This strategy utilizes fibrinogen modified with hydrophobic groups to induce self-assembly into a hydrogel, which is converted into a dry patch. The protein patch enhances adhesion performance on the wet tissue through a dry cross-linking method and robust intra/inter-molecular interactions. This patch demonstrates excellent haemostatic efficacy in both porcine oozing wound and porcine severe acute haemorrhage. It maintains biological functionality, and ensures sustained wound sealing while gradually degrading in vivo, making it a promising candidate for clinical tissue sealing applications.

Uncontrolled haemorrhage accounts for over 30% of trauma-related fatalities. Rapidly achieving definitive haemostasis is essential for alleviating patient discomfort and enhancing surgical efficiency[1–3]. Existing strategies employing procoagulant materials, such as thrombin and zeolite, are limited by slow coagulation and weak mechanical strength[4,5]. Other approaches involve the use of bioadhesives that provide physical barriers to seal the bleeding sites[6–11]. The clinical bioadhesives, such as BioGlue® and cyanoacrylate, have suffered either poor adhesion ability or potential biological toxicity in a moist physiological environment[12]. Extensive research efforts have led to the development of bioadhesives capable of repelling the interfacial water and forming robust adhesion on the tissue surface[13,14]. Nevertheless, the current high-performance bioadhesives primarily consist of synthetic polymer hydrogels, which exhibit absence of degradability in vivo or insufficient biocompatibility, limiting their further clinical translation[15]. Hence, the adhesion efficacy, biocompatibility and biodegradability of bioadhesives are all of great clinical significance.

[1]Department of Hepatobiliary and Pancreatic Surgery, The Second Affiliated Hospital, School of Medicine, Zhejiang University, Zhejiang Hangzhou 310009, China. [2]Key Laboratory of Precision Diagnosis and Treatment for Hepatobiliary and Pancreatic Tumor of Zhejiang Province, Zhejiang Hangzhou 310009, China. [3]MOE Key Laboratory of Macromolecular Synthesis and Functionalization, Department of Polymer Science and Engineering, Zhejiang University, Zhejiang Hangzhou 310058, China. [4]Research Center of Diagnosis and Treatment Technology for Hepatocellular Carcinoma of Zhejiang Province, Zhejiang Hangzhou 310009, China. [5]Center for Medical Research and Innovation in Digestive System Tumors, Ministry of Education, Zhejiang Hangzhou 310009, China. [6]Cancer Center, Zhejiang University, Zhejiang Hangzhou 310058, China. [7]State Key Laboratory of Transvascular Implantation Devices, Zhejiang Hangzhou 310009, China. [8]These authors contributed equally: Lisha Yu, Zhaodi Liu. ✉e-mail: wam@zju.edu.cn; dingyuan@zju.edu.cn; zwmao@zju.edu.cn

Protein-based hydrogels provide excellent biocompatibility and biodegradability, which not only significantly minimize the risk of adverse reactions but also ensure controlled absorption in the body[15–20]. Several blood-derived and other naturally derived protein hydrogel adhesives have been developed, but they require extensive pre-treatment or UV light exposure to initiate covalent bond formation[15,16]. The typical protein-based hydrogels adhere insufficiently to wet tissue due to the abundance of water in hydrogels and the interfacial water, and/or modestly promote the healing process[21]. Furthermore, chemical cross-linking methods of proteins can lead to a reduction in biological activity[22–24], and the cross-linking initiators and/ or linkers may introduce potential toxicity concerns[25,26]. Therefore, there are two challenges in developing protein-based hydrogels that exhibit both high performance and excellent biocompatibility: (i) minimizing the interference from body fluid; and (ii) ensuring that the bioactive proteins can form hydrogels while preserving their biological functionality and biocompatibility. Several protein-based patches have been developed to remove body fluid and improve adhesion performance, such as fibrin sealant patches (EVARREST®, TachoSil®)[27]. However, the requirement for prolong application of consistent pressure (at least 3 min) to achieve adhesion, low efficiency of massive haemorrhage control (e.g. major arterial or venous bleeding), and composition-related safety concerns (e.g. thrombin, thrombogenic risks) substantially limit their versatility for clinical applications[11,28].

In this study, we propose a self-assembly hydrogel strategy that involves regulating the intra/inter-molecular interactions between fibrinogen molecules. This regulation is achieved by modifying the fibrinogen with hydrophobic groups via a N-hydroxysuccinimide ester (NHS ester) reaction. The self-assembly protein hydrogel is transformed into a dry protein patch, which relies on a dry cross-linking method to minimize the interfacial water and enables rapid cross-linking to the surface, consequently enhancing adhesion strength. The protein patch exhibits efficient haemostatic sealing in the cases of porcine oozing wound (liver injury) and massive acute haemorrhage (femoral artery injury). Moreover, the modified fibrinogen retains its biological functionality, including its ability to interact with thrombin (for blood coagulation) and plasmin (for degradation), while also enhancing the wound healing process. It can maintain wound sealing for up to 2 weeks and is capable of gradually degrading in vivo. The impressive combination of adhesion strength, biocompatibility, biodegradation, and healing capacity makes it a potential candidate for surgical tissue sealing.

## Results

### Preparation and characterizations of FgC6 patch

We achieve strong molecular interactions that induce fibrinogen assembly through a physical mechanism: hydrophobicity-induced molecular entanglement. Fibrinogen (Fg) is a 340-kDa protein composed of two symmetrical halves, each consisting of three polypeptide chains. The Fg molecule exhibits a rod-like structure, with approximately 45 nm in length and 2-5 nm in diameter (Fig. 1a)[29,30]. The hydrophobic groups were conjugated to primary amines on lysine residues of Fg through the NHS ester reaction, leading to a significant alteration in the molecular interaction (Fig. 1a). As an illustration, the hexanoyl group (C6) was conjugated to the lysine residues of Fg (denoted as FgC6). The degree of substitution was determined to be 21.6 ± 4.0% of lysine residues via liquid chromatography-tandem mass spectrometry (LC-MS/MS), corresponding to 1.5 ± 0.3% of amino acid residues on Fg (Supplementary Fig. 1). Remarkably, this minor substitution on FgC6 led to the formation of a self-assembly hydrogel (Fig. 1b). Unmodified Fg remained in a soluble state. The FgC6 self-assembly hydrogel exhibited a storage modulus G′ of up to 71.9 ± 1.7 Pa and a complex viscosity η* exceeding 12.4 Pa·s (Fig. 1c, d). Rheological analysis revealed that the storage modulus G′ of FgC6 surpassed the loss modulus G″, indicating its hydrogel properties. Both the storage

modulus G′ and loss modulus G″ of the FgC6 hydrogel were significantly elevated compared to those of Fg solution. The storage modulus G′ of FgC6 (71.9 ± 1.7 Pa) was 70 times greater than that of Fg (1.3 ± 0.2 Pa). Cryogenic transmission electron microscopy (cryo-TEM) demonstrated the presence of protein fibers formed from the self-assembly of FgC6 molecules (Fig. 1e). In contrast, Fg molecules maintained a relatively uniform dispersion in the solution. The self-assembly of FgC6 molecules was further assessed using synchrotron small-angle X-ray scattering (SAXS). The scattering intensity and slope value in the low-q regime showed a remarkable increase, suggesting a growing aggregation of the hybrid nanofilaments of FgC6 in comparison to Fg (Fig. 1f, g)[31]. Distinct differences between the two samples were observed from 0.03 nm$^{-1}$ to low q regime, indicating a notable variance in aggregation states at scales exceeding 200 nm. Cryo-TEM and SAXS characterizations exhibited consistent findings.

Notably, the self-assembly-induced gelation within FgC6 system occurred under the high-concentration conditions to ensure protein entanglement and molecular interaction, typically at 50 mg/mL and above (Supplementary Fig. 2). Due to minor modification (1.5 ± 0.3% of amino acid residues) and limited protein entanglement at a low concentration (10 mg/mL), FgC6 maintained its capacity to interact with thrombin and its formation of fibrin clots, underscoring the preservation of its bio-functional properties (Fig. 1h and Supplementary Fig. 3). The self-assembly hydrogel was employed to prepare the FgC6 patch using the freeze-drying method (Fig. 1i). In contrast, Fg failed to form a uniform protein patch due to inadequate protein entanglement and intra-molecular interaction.

### Intra/inter-molecular interactions and protein entanglement of FgC6

To reveal the intra/inter-molecular interactions and protein entanglement of FgC6 molecules, we investigated the structural dynamics of FgC6 molecules with different degrees of substitution using molecular dynamics (MD) simulation. As the self-assembly progressed, FgC6 molecules became increasingly compact due to the interactions between the introduced hexanoyl group and inner hydrophobic regions of the FgC6 molecules, as well as with other neighboring FgC6 molecules (Fig. 2a). The reduced value of the distance of center mass, radius of gyrate (Rg), solvent accessible surface area (SASA) further validated the more compact structure, shorter molecular distance, and increased stability of FgC6 molecules, compared to Fg molecules (Fig. 2b–d). The hexanoyl groups played a crucial role in intra/inter-molecular interactions, with hydrophobic interaction serving as the primary driving force in the assembly process. An increase in hydrogen bond and salt bridge was observed between FgC6 molecules, resulting in enhanced structural stability (Fig. 2e–h). The structural stability and intra/inter-molecular interactions of FgC6 molecules were improved with an increase in the substitution degree ranging from 0% to 40%.

To further confirm intra/inter-molecular interactions of FgC6 molecules, we compared two molecules (dimer, Fig. 2a) and four molecules (tetramer, Supplementary Fig. 4). During the MD simulation process, the Fg molecules exhibited a relatively independent distribution, with no significant variation in the distance of center mass in both the dimer and tetramer systems. In contrast, both the dimer and tetramer molecules of FgC6 demonstrated protein entanglement, leading to a notable reduction in the distance of center mass between the FgC6 molecules. Comparison of the tetramer system of FgC6 molecules with the dimer system revealed significantly enhanced structural stability and increased intra/inter-molecular interactions (Supplementary Fig. 5). This suggests that the hydrophobicity-induced molecule entanglement of FgC6 resulted from strong intra/inter-molecular interactions. Moreover, as the number of FgC6 molecules increases, these interactions and structural stability can be further strengthened (Supplementary Fig. 5b–g), ultimately facilitating protein self-assembly.

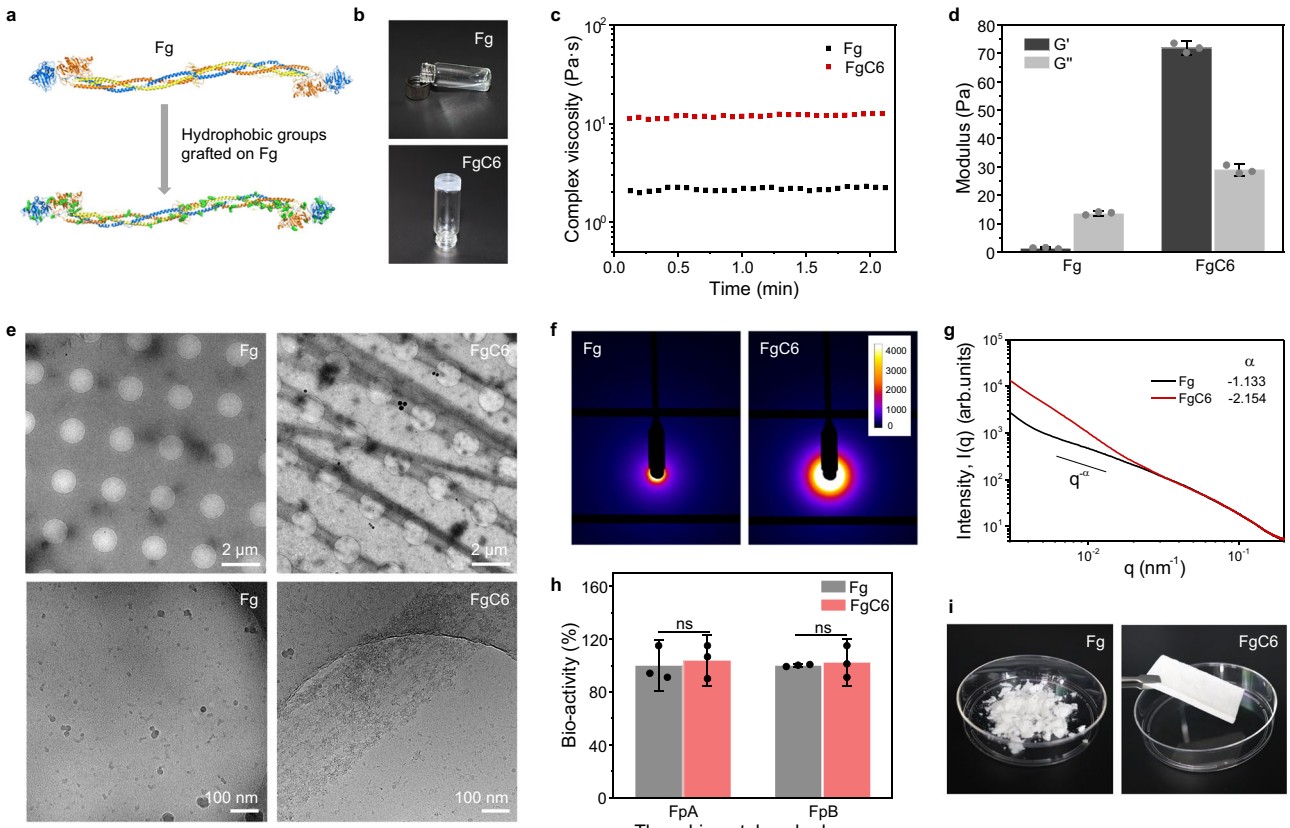

**Fig. 1 | Preparation and characterizations of FgC6 patch. a** Hydrophobic groups were grafted onto lysine residues of Fg molecule, which were demonstrated the protein structure by using PYMOL. The hydrophobic groups were marked with green color. **b** Photograph illustrating the gelation of FgC6 induced by hydrophobic interaction. **c** Complex viscosity corresponding to the Fg and FgC6 systems, respectively. **d** Rheological analysis of Fg and FgC6 systems, respectively ($n = 3$ independent experiments). **e** Low magnification (2600×) and high magnification (73000×) cryogenic transmission electron microscopy (cryo-TEM) images of the Fg and FgC6 systems, respectively. In the FgC6 hydrogel, fibers are observed, formed through protein self-assembly. This contrasts with the homogeneously dispersed protein in the Fg solution. The experiment was repeated two times independently with similar results. **f** Synchrotron small-angle X-ray scattering (SAXS) patterns and **g** curves for Fg solution and FgC6 hydrogel systems. Intensity color bar (arb. units). **h** Fibrinopeptide A (FpA) and fibrinopeptide B (FpB) were cleaved by thrombin from Fg and FgC6 systems at a low concentration (10 mg/mL), respectively ($n = 3$ independent samples). **i** Photograph of Fg powder and FgC6 patch. Error bars, mean ± SD. *P* values are determined by two-sided Student's t-test for **h**; ns: not significant.

An appropriate degree of hexanoyl group (~20%) was selected to analyze the structure of FgC6 molecules. The proportions of the secondary structure evolved with increasing intra/inter-molecular interactions. Analysis of the secondary structure proportions using Fourier transform infrared (FT-IR) spectroscopy indicated that FgC6 molecules displayed a higher β-sheet content and lower levels of other secondary structures, compared to Fg molecules (Fig. 2i, j). The increased formation of β-sheet contributed to the enhanced mechanical strength[32,33]. Both the protein structure and intra/inter-molecular interactions play pivotal roles in enhancing the mechanical performance of the FgC6 hydrogel patch.

**Mechanical property and adhesion performance of FgC6 patch**
The FgC6 patch can rapidly and robustly adhere to wet tissues through a dry cross-linking mechanism (Fig. 3a)[10,34]. The dry FgC6 patch can absorb the interfacial water from the tissue, transforming into a hydrogel that establishes a tight contact surface with wet tissue. Simultaneously, the strong intra/inter-molecular interactions of FgC6 molecules facilitate robust adhesion to the tissue surface. We anticipate that this FgC6 patch can provide a strong and durable haemostatic seal for injured sites. We evaluated the rate and capacity of interfacial water absorption for Fg and FgC6 systems (Fig. 3b). FgC6 demonstrated a slightly faster water absorption rate, while both Fg and FgC6 exhibited similar capacities for absorbing interfacial water. The strong intra/inter-molecular interactions of FgC6 molecules can

enhance its mechanical property. A specific volume of water was dropped onto the surface of FgC6 patch to initiate the hydration process, followed by rheometer analysis to evaluate its mechanical characteristics (Fig. 3c, d and Supplementary Fig. 6). Both the storage modulus (G′) and loss modulus (G″) of the FgC6 patch increased during the hydration process. The final storage modulus (G′) of the FgC6 hydrogel transformed from FgC6 patch was measured at 215.6 ± 17.2 kPa, with loss modulus (G″) of 76.2 ± 2.8 kPa. Remarkably, the final storage modulus (G′) of the FgC6 system was nearly 27 times that of the Fg system. Consequently, the FgC6 patch demonstrated superior mechanical properties, ensuring exceptional cohesion and efficient haemostatic sealing.

The adhesion properties of FgC6 patch were assessed by tensile strength and shear strength based on the American Society for Testing Material (ASTM) standards[11,35]. FgC6 patch exhibited excellent attachment upon contact with blood (tensile strength of 57.4 ± 8.4 kPa), and robust adhesion (shear strength of 31.0 ± 1.6 kPa) to blood-covered porcine skin, which were considerably higher than those of Fg (tensile strength of 6.2 ± 0.6 kPa and shear strength of 3.3 ± 0.1 kPa, Fig. 3e–j), respectively. Meanwhile, in comparison to existing available tissue adhesives, the FgC6 patch showed superior adhesion capabilities. It significantly surpassed the performance of fibrin-based sealants (Fibrin Glue), fibrin-based patches (EVARREST®, TachoSil®), gelatin-based haemostatic matrixes (Surgiflo®(Thrombin)). These commercially available tissue adhesives exhibited low attachment upon

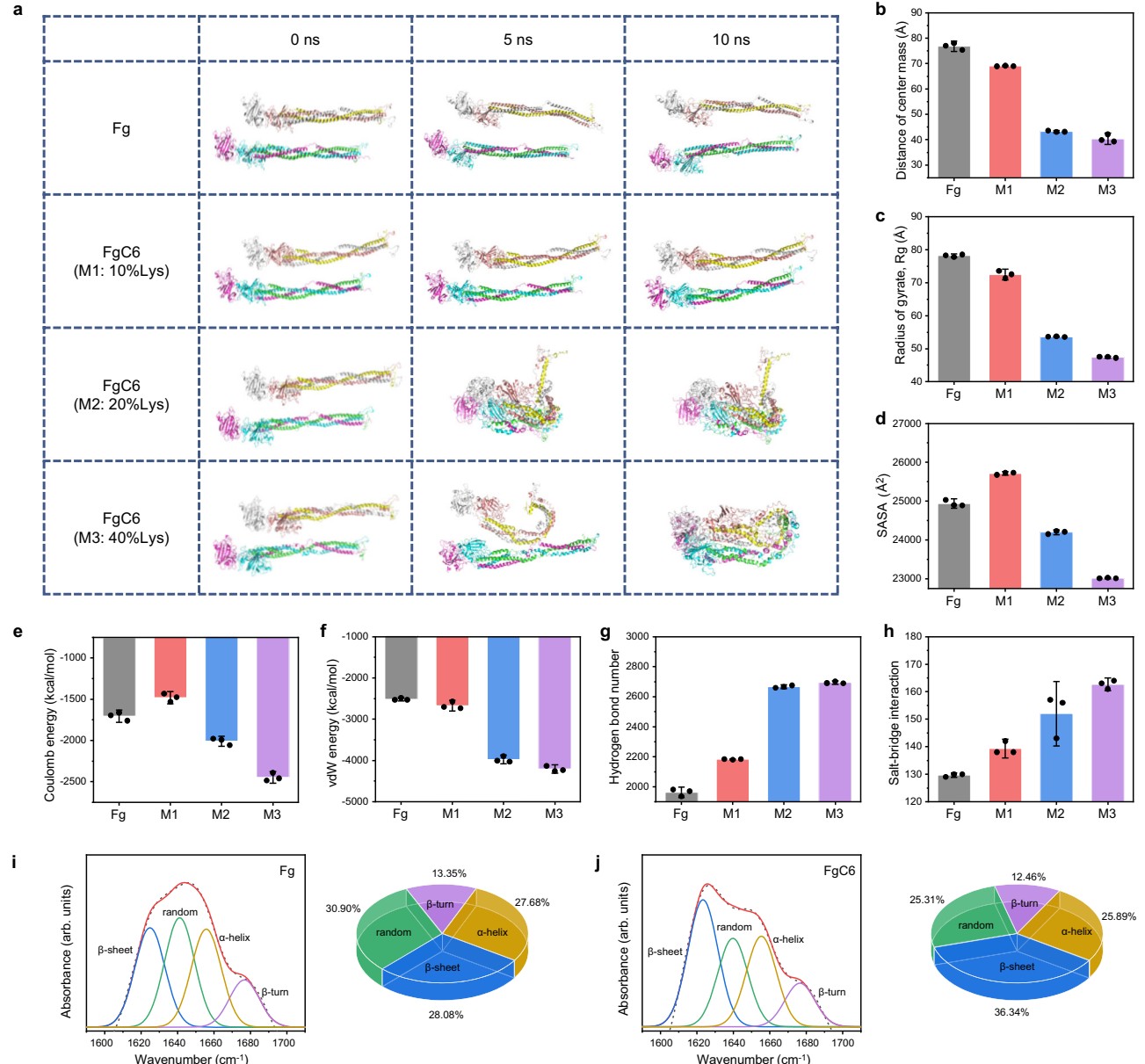

**Fig. 2 | Intra/inter-molecular interactions and protein entanglement of FgC6 molecules resulted in protein self-assembly. a** Structural dynamic of FgC6 molecules with different degrees of modification during molecular dynamics (MD) simulation in 10 ns. Modification 1 (M1): 10% substitution degree of lysine (10%Lys). Modification 2 (M2): 20% substitution degree of lysine (20%Lys). Modification 3 (M3): 40% substitution degree of lysine (40%Lys). **b–h** Distance of center mass, radius of gyrate (Rg), solvent accessible surface area (SASA), Coulomb energy, vdW energy, hydrogen bond number, and salt-bridge interaction in the Fg and FgC6 systems, respectively (*n* = 3 points at energy equilibration). **i, j** Relative content of the secondary structure in Fg and FgC6 systems analyzed using Fourier transform infrared (FT-IR) spectroscopy. Error bars, mean ± SD.

contact with blood (tensile strength less than 8 kPa), and weak adhesion (shear strength less than 10 kPa) to blood-covered porcine skin. Furthermore, FgC6 patch offered excellent adhesion capacity compared to other commercial haemostatic materials (oxidized cellulose based-dressing, Surgicel® Fibrillar; gelatin-based dressing, Gelatin Sponge; Supplementary Fig. 7).

In view of the intra/inter-molecular interactions of FgC6 molecules in improving adhesion performance, we further explored the universality of hydrophobic groups in the construction of self-assembly hydrogel patch, in which acetyl (C2), propionyl (C3), hexanoyl (C6), decanoyl (C10), dodecanoyl (C12), benzoyl (Ben), 2-(naphthalene-1-yl)acetyl (Nap) were selected as representative hydrophobic groups, respectively (Supplementary Tables 1-3). Initially, it has been demonstrated that structural stability and intra/inter-molecular

interactions of FgC6 were enhanced with an increasing substitution degree ranging from 0% to 40% in MD simulation (Fig. 2b–h). The shear strength of the self-assembly protein patches on the porcine skin also presented an enhanced tendency with the increase of the substitution degree (Supplementary Table 1-2). Besides, as listed in Supplementary Table 3, various hydrophobic groups, such as C2, C3, C6, C10, C12, Ben, Nap, and so on, can strengthen the intra/inter-molecular interactions of protein, consequently improving the adhesion performance to the wet tissue. Conversely, hydrophilic groups, including 10-oxocapric acid acyl (C9COOH) and 2,5,8,11-tetraoxatetradecane-14-acyl (C10O4), exhibited either comparable or lower adhesion strength when compared to Fg system (Supplementary Table 3). This observation indicates that hydrophilic groups do not enhance intra/inter-molecular interaction nor promote protein entanglement. The self-assembly

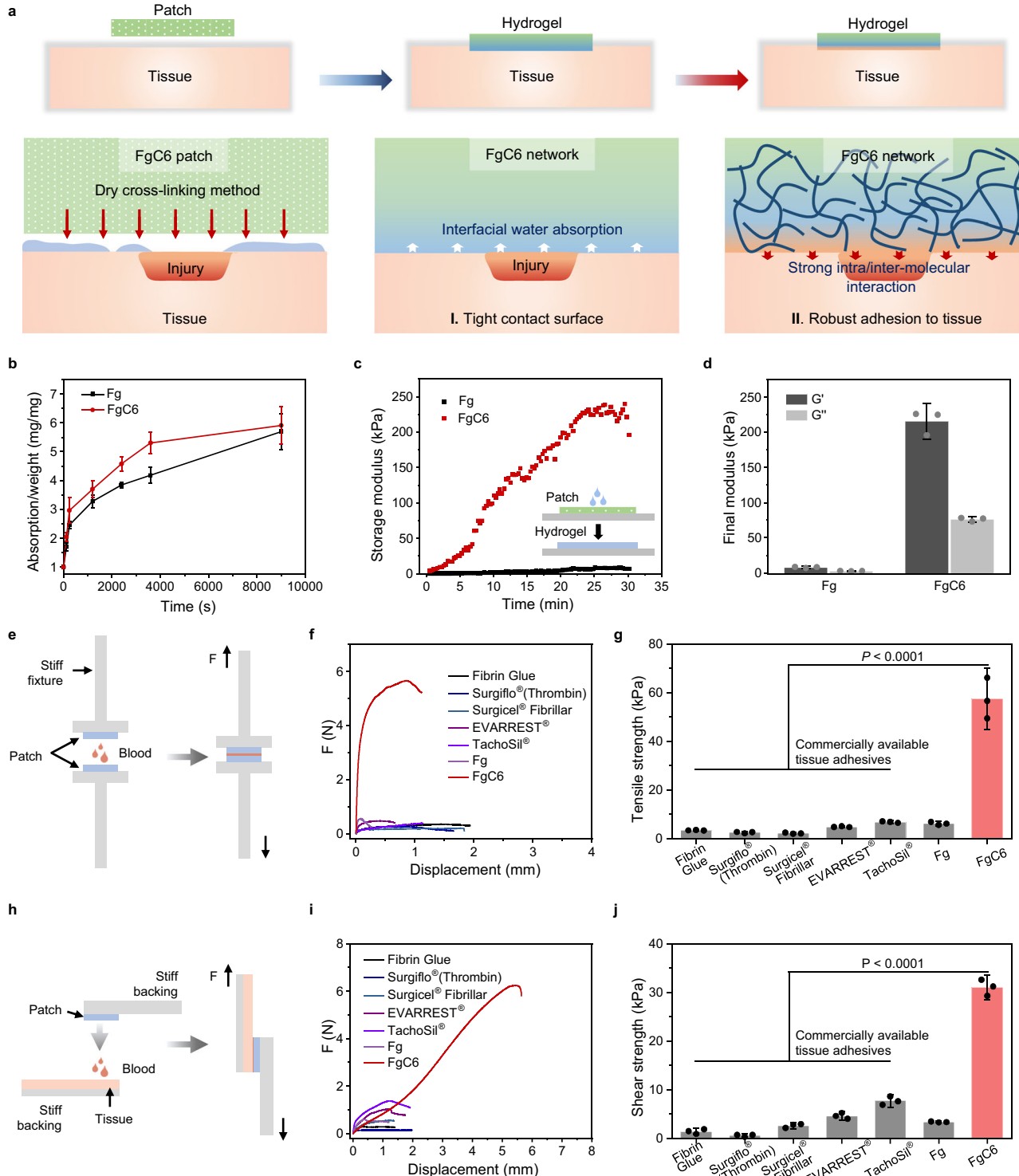

**Fig. 3 | Mechanical property and adhesion performance of FgC6 patch.**
**a** Schematic illustration for tight contact surface via interfacial water absorption, and robust adhesion to tissue via strong intra/inter-molecular interaction. **b** Water absorption kinetic of Fg and FgC6 systems (n = 4 independent experiments). **c** Transformation of Fg powder and FgC6 patch systems into the hydrogel or solution state using rheological analysis through dynamic time-sweep tests. **d** Final storage modulus (G') and loss modulus (G") of Fg solution and FgC6 hydrogel transformed from Fg powder or FgC6 patch, respectively (n = 3 independent experiments). **e** Schematic of tensile test used for determining tensile strength based on the modified ASTM F2258-05 standard. **f** Force–displacement curves of

the tensile test. **g** Tensile strength of Fibrin Glue, Surgiflo®(Thrombin), Surgicel® Fibrillar, EVARREST®, TachoSil®, Fg, and FgC6, respectively (n = 3 independent samples). **h** Schematic illustrating the lap shear test used for assessing the shear strength on one side of porcine skin based on the modified ASTM F2255-05 standard. **i** Force–displacement curves of the lap shear test. **j** Shear strength of Fibrin Glue, Surgiflo®(Thrombin), Surgicel® Fibrillar, EVARREST®, TachoSil®, Fg, and FgC6 on one side of porcine skin (n = 3 independent samples). Error bars, mean ± SD. *P* values are determined via one-way ANOVA followed by Tukey's post-hoc test for **g** and **j**.

hydrogel patch induced from protein entanglement belongs to physical cross-linking[36–38]. Three types of fibrinogen-based patches (FgC6, chemical, and enzymatic cross-linking patches) were prepared using the same source of fibrinogen to compare their adhesion performance. The chemical cross-linking of methacryloyl Fg and the enzymatic cross-linking of Fg were prepared to form the referential patches, both of which exhibited significantly lower adhesion performance. We hypothesize that a delicate balance was achieved between cohesion and adhesion through the adjustment of the intra/inter-molecular interactions of FgC6 molecules. This well-balanced protein patch demonstrated robust tissue adhesion strength. Besides, in comparison to chemical and enzymatic cross-linking patches, our method presents potential benefits in terms of cost-effectiveness, simplicity of composition, and preservation of protein bio-functionality.

## Biocompatibility and biodegradability of FgC6 patch

In vitro cytotoxicity assays and an in vivo rat subcutaneous implantation model were utilized to assess the biocompatibility and biodegradability of the FgC6 patch. The FgC6 patch's in vitro cytotoxicity was assessed using CCK-8 and live/dead assays. The FgC6 patch showed equivalent effects on cell viability and proliferation after 1, 3, 5, and 7 days of treatment when compared with the control group (Fig. 4a, b). These results suggest that the FgC6 patch possesses excellent cell compatibility and is potentially safe for future biomedical applications. Fg and fibrin undergo biodegradation facilitated by endogenous enzymes like plasmin, and Fg/fibrin degradation products (FDP) of FgC6 system were examined to evaluate biodegradation activity. The in vitro biodegradation capacity of FgC6 system was observed to be comparable to that of Fg system, ensuring its reliable biodegradability (Fig. 4c).

The FgC6 patch's in vivo biodegradability and biocompatibility were assessed through dorsal subcutaneous implantation in a rat model (Fig. 4d). In the control group, Surgicel® Fibrillar, a haemostatic patch was implanted. During the implantation period, both FgC6 patch and Surgicel® Fibrillar demonstrated a progressive reduction in volume, culminating in complete degradation by the end of 4 weeks (Fig. 4e, f). No signs of necrosis were observed in the muscle or skin layer around the implantation site. To evaluate the potential systemic toxicity of the FgC6 patch, we conducted complete blood counts (CBCs) and comprehensive blood chemistry panels. During the 2-week study duration, blood analysis from animals with the implanted FgC6 patch were similar to those from healthy rats, showing no notable indication of systemic toxicity (Fig. 4h, i). The histological evaluation using hematoxylin and eosin (H&E) results indicated that FgC6 patch elicited an inflammation response in the surrounding tissue comparable to that observed with the commercial product Surgicel® Fibrillar (Fig. 4g).

The biocompatibility and wound healing efficacy of the FgC6 patch was also evaluated by in a rat liver incision injury model (Fig. 5a). A liver injury model was established by creating an incision injury with a length of 8 mm and a depth of 3 mm. FgC6 patch and Surgicel® Fibrillar were applied at the bleeding site until haemorrhage control was achieved, respectively. The FgC6 patch maintained on the injured liver for 4 weeks (Fig. 5b), and it demonstrated a progressive reduction in volume throughout the implantation period (Fig. 5c). After repairing the injured liver, the surface of liver remained smooth, and no polyps or tissue adhesion were observed at 1, 2, 4 weeks post-surgery. This manifested that the FgC6 patch does not lead to postoperative adhesion. H&E images of the liver tissues indicated that the FgC6 patch induced a level of inflammation comparable to that observed with the commercial product Surgicel® Fibrillar at 2 weeks post-surgery. Macrophages are crucial in the wound healing process, which includes inflammation, proliferation, and remodeling stages. Throughout this process, there is a dynamic transition in local macrophage population from a primarily pro-inflammatory (M1) state to a predominantly anti-inflammatory (M2) state. M1 macrophages promote immune response in the initial stage of wound repair, and combat foreign pathogens and bacteria[39,40]. M2 macrophages primarily contribute to tissue repair in the later stage of wound healing[7]. Immunofluorescence staining was performed to visualize pan-macrophages using CD68 marker and M2 type macrophages using CD206 marker. The normalized intensity of the immunofluorescence signal at 2 weeks post-treatment was analyzed to demonstrate the relative abundance of these cell types (Fig. 5e–g). Marker CD68 for pan-macrophages revealed that FgC6 patch and Surgicel® Fibrillar induced comparable levels of inflammation. Immunofluorescence intensity analysis of marker CD206 for M2 type macrophages manifested that FgC6 patch could potentially promote wound healing. Moreover, FgC6 patch on the injured liver exhibited major degradation observed at 4 weeks post-implantation, with no significant inflammation on the repairing liver (Fig. 5d, e). Complete blood counts (CBCs) and comprehensive blood chemistry were carried out to assess the biosafety of the FgC6 patch. During the study period of 4 weeks, blood analysis results from rats with the implanted FgC6 patch were comparable to the results observed in healthy rats (Fig. 5h, i). Indicators of liver parenchymal damage manifested that the animals treated with FgC6 patch exhibited relatively normal liver function. Further in-depth quantitative assessment of FgC6 patch on inflammation regulation and wound healing needs to be evaluated prior to its clinical applications. This includes investigating the dense structure formed by high concentrations of FgC6 and its impact on the wound-healing process. The promising long-term efficacy, biocompatibility, and healing ability of the FgC6 patch make it a potential candidate for surgical tissue sealing.

## Haemostatic sealing in porcine bleeding models

We demonstrated the rapid haemostatic sealing capabilities of bleeding injuries in pigs, validating the in vivo efficacy of FgC6 patch in a setting that is representative of clinical applications. In the liver resection injury model, a part of the liver lobe, specifically about 50 mm in length was cut off[41,42]. And free bleeding was allowed for 30 s (pretreatment blood loss) to assess bleeding rates (Supplementary Table 4). Commercially available haemostatic materials, including Surgicel® Fibrillar, Surgiflo®(Thrombin), and TachoSil® were applied to bleeding liver injury, following the manufacturer's guidelines (Supplementary Movies 1–3). Due to strong adhesion performance, FgC6 achieved haemostatic sealing after 30 s of manual compression, showing a significantly reduced haemostatic time and blood loss, compared to Surgicel® Fibrillar, Surgiflo®(Thrombin) and TachoSil® (Fig. 6a–c and Supplementary Movie 4).

Femoral artery bleeding injury is an acute haemorrhage that requires a short period of time to control, otherwise it can be life-threatening[43]. High-pressure and high-volume bleeding from arterial injuries require strong adhesion or effective procoagulant performance to effectively seal the injury. An injured defect with 6 mm of incision was made on the femoral artery to establish the porcine femoral artery bleeding injury[44,45]. And unrestricted bleeding was allowed for 45 s (pretreatment blood loss) to assess bleeding rates (Supplementary Table 5). Combat Gauze®, which is recommended for massive haemorrhage in Tactical Combat Casualty Care (TCCC) Guidelines[46], was used as a reference group (Supplementary Table 6). Combat Gauze® needs to be applied with at least 3 min of direct pressure to control bleeding. In the porcine femoral artery bleeding injury, it took Combat Gauze® over 4 min to achieve haemostasis (Fig. 6d, e and Supplementary Movie 5). In contrast, FgC6 patch can induce strong seal of the artery defect within 1 min, restoring the normal bloodstream immediately (Fig. 6d, and Supplementary Movie 6). Rapid and strong haemostatic sealing obviously reduced the posttreatment blood loss. Thus, the FgC6 patch group experienced a blood loss of $0.4 \pm 0.2$ g, which was approximately 98% less than the blood loss in Combat Gauze® group (Fig. 6e, f).

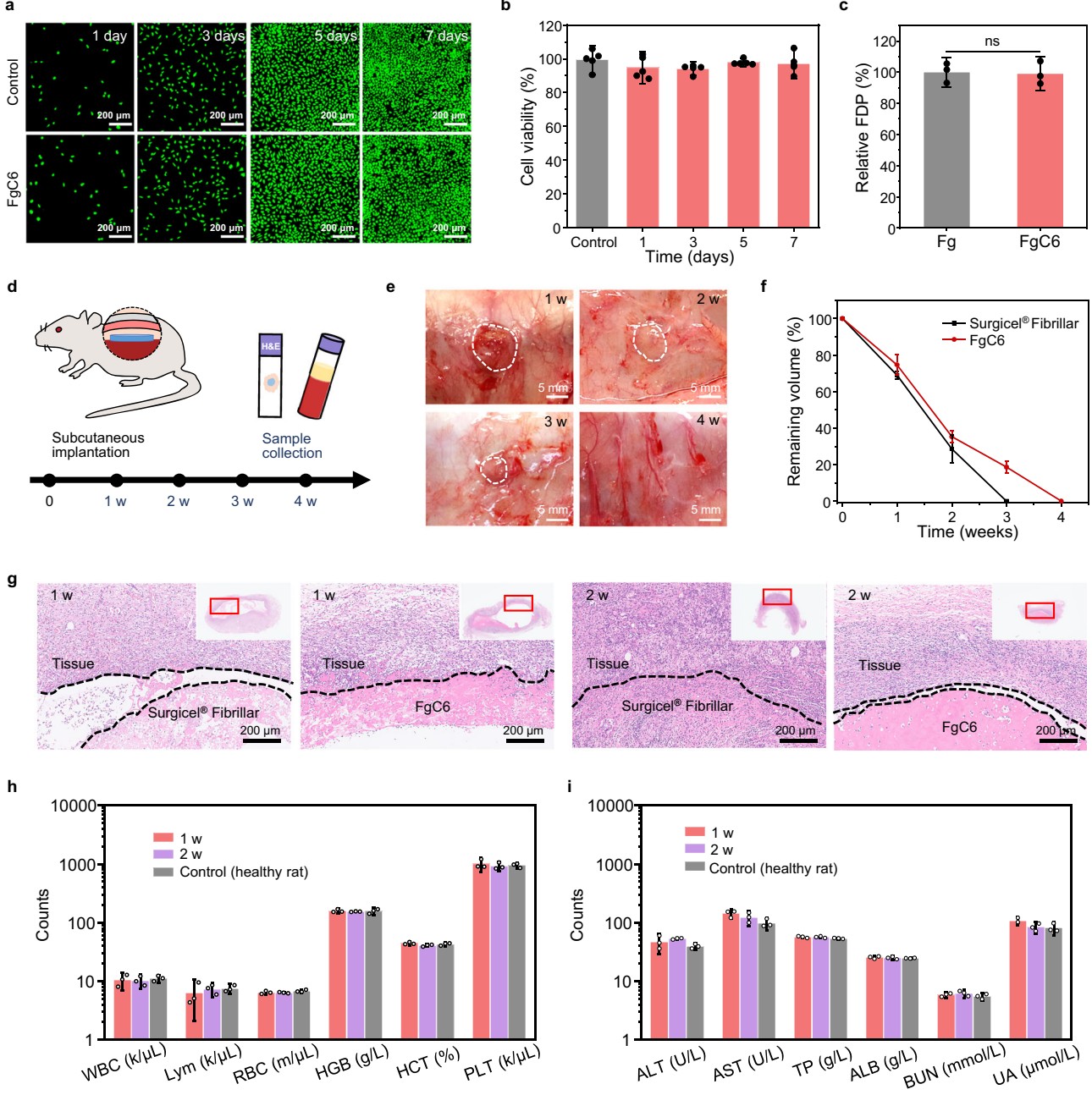

**Fig. 4 | Biocompatibility and biodegradability of FgC6 patch. a** Live/dead staining of L929 cells cultured in the control media and FgC6 patch-incubated media for durations of 1, 3, 5, and 7 days. Calcein acetoxymethyl was used to stain the living L929 cells (green), and propidium iodide was used to identify dead L929 cells (red). The experiment was repeated five times independently with similar results. **b** Cytotoxic effects of FgC6 patch on L929 cells evaluated following incubation periods of 1, 3, 5, and 7 days ($n = 5$ biologically independent replicates). **c** In vitro degradation of Fg and FgC6 systems in phosphate-buffered saline (PBS) with plasmin. The Fg and FgC6 systems can be recognized and degraded by plasmin, and fibrin/fibrinogen degradation products (FDP) were investigated ($n = 3$ independent samples). **d** Schematic of the dorsal subcutaneous implantation in a rat model including the subsequent analysis of samples. **e** Image of the tissue with subcutaneously implanted FgC6 patch after implantation. **f** Degradation rate of

Surgicel® Fibrillar and FgC6 patch after subcutaneous implantation ($n = 3$ biologically independent replicates). **g** Representative histological images using hematoxylin and eosin (H&E) for the surrounding tissue after subcutaneous implantation of Surgicel® Fibrillar and FgC6 patch at 1, 2 weeks. The experiment was repeated three times independently with similar results. **h** Complete blood counts (CBCs) of control (healthy rats) and rats with the FgC6 patch subcutaneously implanted at 1, 2 weeks. White blood cell: WBC; lymphocyte: Lym; red blood cell: RBC; hemoglobin: HGB; hematocrit: HCT; platelet: PLT ($n = 3$ independent rats). **i** Blood chemistry of control (healthy rats) and rats with the FgC6 patch subcutaneously implanted at 1 and 2 weeks. Alanine transaminase: ALT; aspartate transaminase: AST; total protein: TP; albumin: ALB; blood urea nitrogen: BUN; Urea: UA ($n = 3$ independent rats). Error bars, mean ± SD. *P* values are determined by two-sided Student's t-test for **c**. ns: not significant.

Therefore, the FgC6 patch can achieve haemostatic sealing in both oozing wound (such as liver injury) and massive acute haemorrhage (such as femoral artery injury). This is facilitated by the removal of interfacial water and the strong intra/inter-molecular interactions of

FgC6 molecules. Meanwhile, the FgC6 patch exhibited stable sealing ability and excellent biocompatibility.

In summary, the study presents a significant advancement in the development of a protein-based hydrogel patch through a molecular

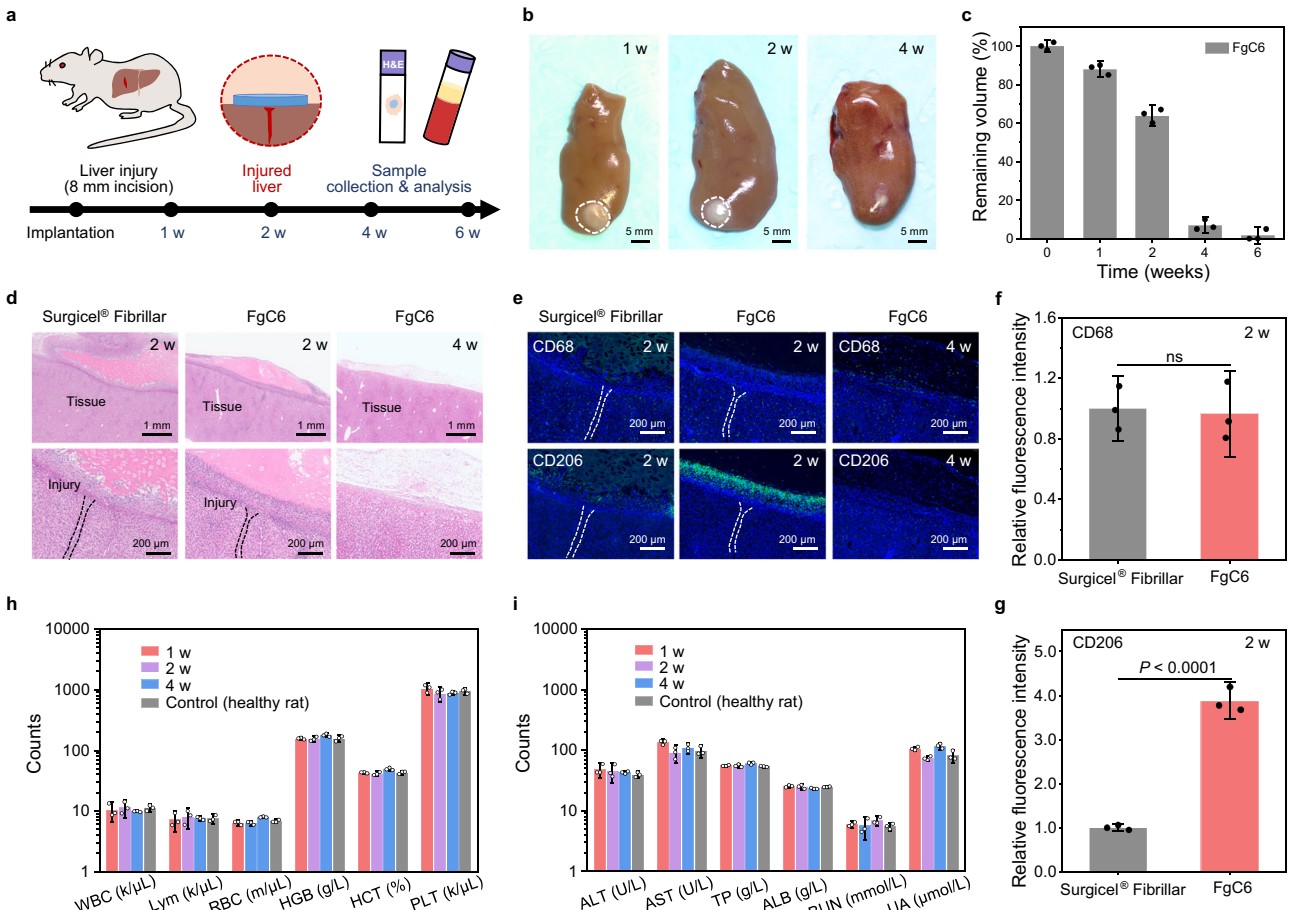

**Fig. 5 | Biocompatibility and wound healing efficacy of FgC6 patch. a** Schematic of implanting FgC6 patch on the liver incision injury in the rat model. **b** Rat liver collected at 1, 2, 4 weeks post-sealing with the FgC6 patch. **c** Degradation rate of subcutaneously implanted FgC6 patch ($n = 3$ biologically independent replicates). **d** Representative histology images stained with H&E for Surgicel® Fibrillar at 2 weeks, and FgC6 patch at 2, 4 weeks after haemostatic treatment. The experiment was repeated three times independently with similar results for **d**. **e** Representative immunofluorescence images for Surgicel® Fibrillar at 2 weeks, and FgC6 patch at 2, 4 weeks after haemostatic treatment. The experiment was repeated three times independently with similar results for **e**. Fluorescence intensity was measured from the immunofluorescence images of the Surgicel® Fibrillar and FgC6 patch group for **f** CD68 ($n = 3$ biologically independent replicates) and **g** CD206 makers ($n = 3$ biologically independent replicates). Cell nuclei were stained with DAPI (blue). Green fluorescence represented the expression of CD68 or CD206. **h** Complete blood counts (CBCs) of control (healthy rats) and rats with the implanted FgC6 patch at 1, 2, 4 weeks ($n = 3$ biologically independent rats). **i** Blood chemistry of control group (healthy rats) and FgC6 patch group at 1, 2, 4 weeks ($n = 3$ biologically independent rats). Error bars, mean ± SD. $P$ values are determined by a two-sided Student's t-test for **f** and **g**. ns, not significant.

self-assembly strategy. The self-assembly protein patch made from hydrophobic group modified fibrinogen, absorbs the interfacial water from the tissue and demonstrates strong intra/inter-molecular interactions of protein molecules, facilitating robust adhesion to the tissue surface. The protein patch exhibits rapid and effective haemostatic sealing in porcine bleeding models, including liver injuries and arterial injuries. It provides strong adhesion and promotes wound healing without causing significant inflammation or toxicity. The biocompatibility, biodegradation, preservation of biological functionality and ease of implementation make it a promising candidate for clinical applications. This work represents an excellent approach to improve trauma care and surgical procedures.

## Methods
### Materials
Bovine fibrinogen (340 kDa) was obtained from Yeasen Biotechnology (Shanghai) Co., Ltd. Human fibrinogen (≥80% of protein is clottable; 340 kDa) was purchased from Sigma-Aldrich. N-(Propionyloxy)succinimide was purchased from Sigma-Aldrich. 2,5-Dioxopyrrolidin-1-yl propionate, N-(hexanoyloxy)succinimide, 2,5-dioxopyrrolidin-1-yl decanoate, 2,5-dioxopyrrolidin-1-yl

dodecanoate were purchased from Bidepharm. Dimethyl sulfoxide was purchased from Sinopharm Chemical Reagent Co., Ltd. Bovine and human thrombin was obtained from Shanghai Boatman Biotechnology Co., Ltd. (Shanghai, China). Fibrin Glue (Shanghai RAAS®), Surgiflo®(Thrombin), EVARREST®, TachoSil®, Combat Gauze®, ChitoSAM®, Surgicel® Fibrillar, and Gelatin Sponge (Xiangen Medical) were obtained from commercial sources. In this study, bovine fibrinogen was mainly utilized to prepare the FgC6 patch. We conducted a comprehensive evaluation of bovine FgC6 patch, including detailed characterizations, mechanical performance, adhesion capability, biocompatibility, biodegradability and haemostatic effectiveness. Due to the unavailability of complete 3D structural elucidation and FpA/FpB/FDP ELISA assays for bovine fibrinogen, human fibrinogen was utilized for the MD simulation, LC-MS/MS analysis, and FpA/FpB/FDP ELISA assay.

### Synthesis of FgC6 patch
Fibrinogen solution was prepared in phosphate-buffered saline with a concentration of 50 mg/mL. N-(hexanoyloxy)succinimide was dissolved in dimethyl sulfoxide with a concentration of 0.4 M,

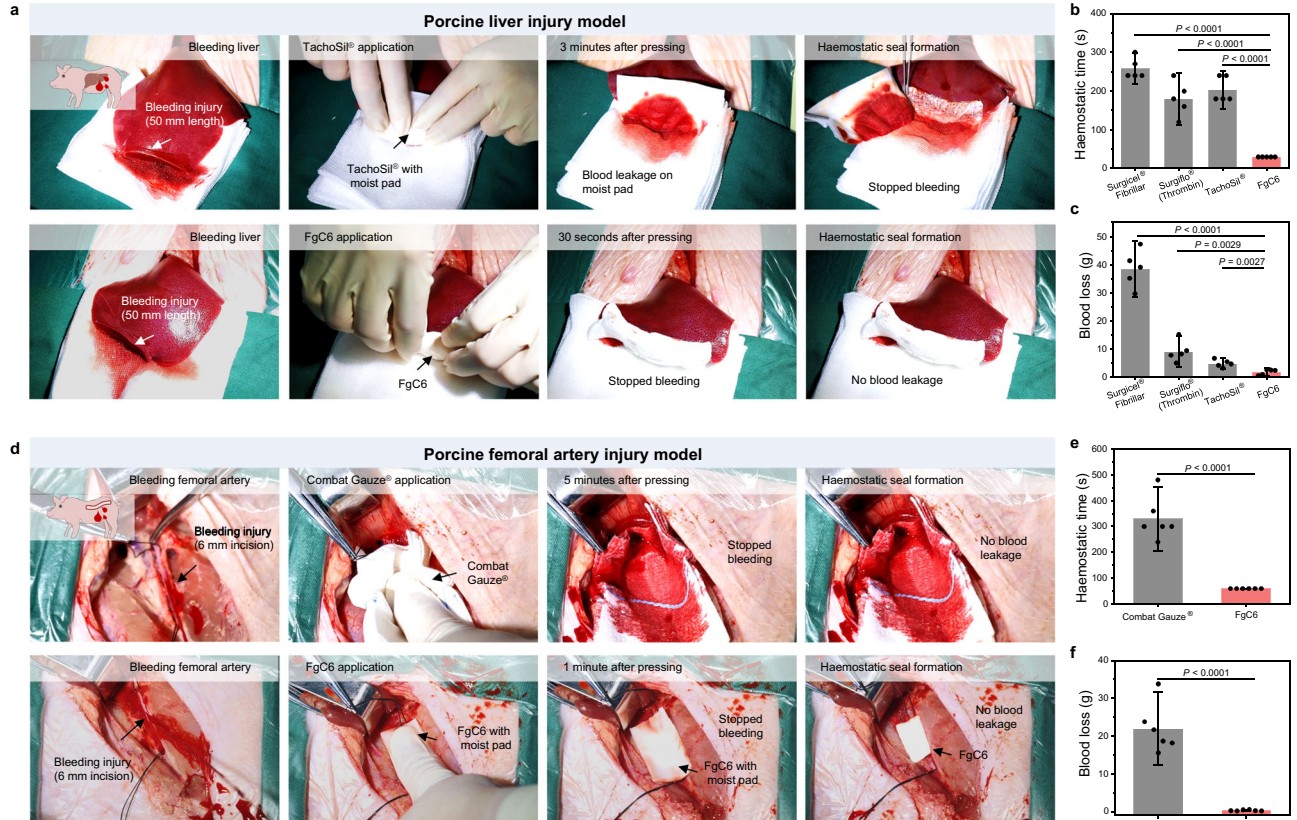

**Fig. 6 | In vivo haemostatic sealing in porcine injury bleeding model.**
**a** Haemostatic treatment in a porcine liver bleeding model. A part of the liver lobe (ca. 50 mm in length) was cut off to establish the porcine liver bleeding model. TachoSil® and FgC6 patch were applied on the wound with manual compression until haemostasis achieved. **b** Haemostatic time and **c** blood loss following treatment of Surgicel® Fibrillar, Surgiflo®(Thrombin), TachoSil® and FgC6 patch in the porcine liver bleeding model ($n = 5$ independent injuries). **d** Haemostatic treatment

in a porcine femoral artery bleeding model. A severe arterial haemorrhage was produced with 6 mm arteriotomy. Combat Gauze® and FgC6 patch were applied on the injured femoral artery with manual compression until haemostasis achieved. **e** Haemostatic time and **f** blood loss following treatment in the porcine femoral artery bleeding model ($n = 6$ independent injuries). Error bars, mean ± SD. $P$ values are determined by two-sided Student's t-test for **b**, **c**, **e** and **f**.

and then was added into the fibrinogen solution. The reaction was conducted at 25 °C for 4 h. The degree of hydrophobic substituent was adjusted by the molar ratios of N-(hexanoyloxy) succinimide to fibrinogen (molar ratio: 14, 27, 54, 82). The solution was dialyzed with phosphate-buffered saline (molecular weight cutoff: 35,000) at 25 °C for 2 days to remove the excess N-(hexanoyloxy)succinimide. After dialysis, the solution was placed in a freezer at −80 °C for a period of 12 h, followed by lyophilization. The resulting product was then stored at 4 °C in the absence of light. The synthesized FgC6 patch (molar ratio of N-(hexanoyloxy)succinimide to fibrinogen: 54) was used for mechanical tests, biocompatibility, and haemostatic performance evaluation.

## Characterizations
**Cryogenic transmission electron microscopy (Cryo-TEM).** Protein entanglement of FgC6 molecules was observed by Cryo-TEM (200 kV, FEI Talos F200C). Vitrified films were prepared by applying 2 μL of the sample on Lacey grids, followed by blotting the sample for 5 s with the blotting force of 3 in a 'Vitrobot' instrument (FEI Company). Low magnification (2600×) and high magnification (73000×) cryo-TEM images were captured for the Fg and FgC6 systems, respectively.

**Synchrotron small-angle X-ray scattering (SAXS).** SAXS measurements were conducted on the BL19U2 beamline at Shanghai Synchrotron Radiation Facility. A 2D detector (Pilatus 1 M with a resolution of 981 × 1043 pixels and a pixel size of 172 μm, Dectris Co. Ltd.) was

utilized to record the data. The X-ray wavelength and sample-to-detector distance were 1.03 Å and 5756.43 mm, respectively. The Fg solution and FgC6 hydrogel system (50 mg/mL) was measured at 5 °C. The acquired raw data were analyzed by FIT2D (version 10.132).

**Fourier transform infrared spectroscopy (FT-IR).** The secondary structure of Fg and FgC6 systems was characterized by FT-IR spectroscopy (Thermo Scientific Nicolet iS20). The resolution was 4 cm$^{-1}$, and the scan number was 32 times.

**Rheology.** Rheological measurements were carried out using a rotary rheometer (HAAKE RS6000) equipped with parallel plate geometry. Complex viscosity, storage modulus G′ and loss modulus G″ of Fg solution and FgC6 hydrogel were monitored at a frequency of 1 Hz with a strain of 1% for the time sweep. Each group contains three independent samples.

**Molecular dynamics (MD) simulation**
The human fibrinogen structure (3GHG) was obtained from the Protein Data Bank (PDB) database. Due to its large molecular weight, one symmetrical half containing three polypeptide chains was selected for molecular dynamics (MD) simulation. The side chain of lysine residue was modified to -NH-CO-(CH$_2$)$_4$CH$_3$ to obtain FgC6 molecule. The substitution degree was 0%, 10%, 20%, 40%, respectively. And 20 % substitution degree of FgC6 was utilized to compare the interactions between the dimer and tetramer systems. UCSF Chimera was used to remove water molecules as well as unrelated heteroatoms, leaving only

the protein structure. To characterize the intra/inter-molecular interaction and mechanical properties, MD simulation was performed in GROMACS 5.1.5 software package. Each system was simulated in a closed dodecahedral periodic boundary environment, without any solvent model. The periodic boundary setting of the simulated system was centered on the protein, and the minimum distance between the edge of the protein and the box was set to 0.1 nm. The AMBEff99SB force field was used for protein-protein interaction simulation. After the initial system was built, the steepest descent method was employed for all atoms to minimize the system's energy. All simulation systems were performed in the constant number of particles, volume, and temperature (NVT) for 1000 ps and the constant number of particles, pressure and temperature (NPT) for 1000 ps. The temperature was controlled at 298.15 K, and the pressure was controlled at 1 bar. After NVP and NPT equilibrium, simulation was carried out with a time step of 2 fs. The long-range electrostatic interaction was calculated by Particle Mesh Ewald (PME) method. Each group contains three points at energy equilibration. PyMOL and Origin software was used to visualize simulation results.

### Interfacial water absorption

The swelling behavior of prepared samples were assessed by placing them on the surface of a water reservoir[47]. The duration of contact was set from 1 to 150 min at room temperature. The degree of water absorption was calculated using the formula $\frac{m_1 - m_0}{m_0}$, where $m_1$ is the mass of measured sample after water absorption, and $m_0$ is the original mass of the dry sample. Each group contained four independent experiments.

### Mechanical tests

Rheological property was performed using a rotary rheometer (HAAKE RS6000) equipped with parallel plate geometry, specifically utilizing a P20 TiL plate with a 20-mm diameter. The as-prepared samples with a diameter of 20 mm and a thickness of 2.5 mm (50 mg of protein) were placed on the parallel plate for testing. 100 μL of the deionized water was added to the upper surface of the samples at 37 °C, and the protein concentration was about 50 mg/100 μL (500 mg/mL). After 10 min, a time-sweep oscillatory test was conducted at a strain of 5% and a frequency of 1 Hz at 37 °C. The storage modulus G′ and loss modulus G″ were recorded. Each group contained three independent samples.

The tensile test was performed using the modified ASTM F2258-05 standard. One side of each of the two prepared samples (10 mm × 10 mm) were fixed on the polytetrafluoroethylene fixtures using the cyanoacrylate glue, respectively. Then, 40 μL of blood was added between the two prepared samples. After 10 min, the polytetrafluoroethylene fixtures with the prepared samples were positioned into a mechanical tester (Zwick/Roell Z020). The samples were subjected to a tensile test at a constant speed of 2 mm/min. The tensile strength was calculated by dividing the maximum force by the area of adhesion. Each group contained three independent samples.

The lap shear test was performed using the modified ASTM F2255-05 standard. One side of the prepared sample (20 mm × 10 mm) was fixed on the stiff backing (polyethylene glycol terephthalate film) using the cyanoacrylate glue. One side of fresh porcine skin was also fixed on the stiff backing (polyethylene glycol terephthalate film) using the cyanoacrylate glue. Then, 80 μL of blood was added to another side of the porcine skin. The prepared sample was added upon the blood-covered porcine skin. After 10 min, the porcine skin attached to the prepared samples was placed into a mechanical tester (Zwick/Roell Z020). The samples were then subjected to a tensile test at a constant speed of 5 mm/min. The shear strength on the porcine skin was calculated by dividing the maximum force at the point of detachment by the area of adhesion. Each group contained three independent samples.

### FgC6 activity identified by thrombin (the activity for forming fibrin clot)

**Fibrinopeptide A/fibrinopeptide B (FpA/FpB) release.** FgC6 or Fg (10 mg/mL, 400 μL) was incubated with 40 μL of thrombin (1 kU/mL) and 40 μL of CaCl$_2$ solution (0.2 M) on a thermoshaker at 37 °C for 8 min. The FpA and FpB in the supernatant was assessed via FpA or FpB ELISA assay (Hefei Laier Biotechnology Co., Ltd.). Each group contained three independent samples.

**Fibrin crosslinking time.** FgC6 or Fg at a concentration of 10 mg/mL was combined with varying concentrations of thrombin (0.5-20 U/mL) at 37 °C. The criterion of effective fibrin crosslinking was the formation of a visible white clot that adhered to the wall of a polystyrene tube. The time taken for fibrin crosslinking was measured and recorded. Each group contained three independent samples.

### FgC6 activity identified by plasmin (the activity for biodegradation)

FgC6 or Fg (10 mg/mL) was mixed with thrombin (100 U/mL) at 37 °C to initiate fibrin clot formation. Subsequently, the mixture containing fibrin clot was treated with plasmin (25 μg/mL) at 37 °C, leading to the degradation into fibrin/fibrinogen degradation products (FDP). The FDP in the supernatant of FgC6 and Fg systems was evaluated via FDP ELISA assay (Hefei Laier Biotechnology Co., Ltd.). Each group contained three independent samples.

### In vitro biocompatibility evaluation

The cytotoxicity test was conducted using the patch-conditioned medium for cell culture. To prepare the patch-conditioned medium, the sterilized patch (10 mg/mL) was incubated in DMEM supplemented with 10 v/v% fetal bovine serum at 37 °C with a shaking speed of 100 rpm for 24 h. L929 cells (ATCC CCL-1) were seeded in 96-well plates at a density of 1000 cells/well, and pre-cultured for 24 h at 37 °C in a 5% CO$_2$ humidified incubator. L929 cells are routinely authenticated by cellular morphology assay. The DMEM was used as a control. Then, the cells were treated with sterilized patch-incubated medium, and further incubated for 1, 3, 5, and 7 days ($n = 5$ biologically independent replicates). The cell viability was evaluated by Cell Counting Kit-8 (CCK-8, Solarbio) method. The absorbance at 450 nm was measured. The cell viability was further determined using live/dead assay (Beyotime Biotechnology). Fluorescence microscope (KEYENCE) was used to image live cells at excitation/emission wavelengths of 490 nm/515 nm, and dead cells at 535 nm/617 nm, respectively.

### In vivo biocompatibility and biodegradability

Male Sprague Dawley rats (SD, 8 weeks, 250-300 g) were utilized for in vivo biocompatibility and biodegradability studies. The studies were approved by the Laboratory Animal Welfare and Ethics Committee of Zhejiang University (ZJU20220337). The biocompatibility and biodegradability were assessed through dorsal subcutaneous implantation in a rat model. Both of Surgicel® Fibrillar and FgC6 patches were shaped into disks, measuring 10 mm in diameter and 3 mm in thickness. Rats were anesthetized using 2% sodium pentobarbital. A 1-2 cm skin incision was made on the rat's back for each implant to access the subcutaneous space. Surgicel® Fibrillar and prepared patch were placed in the subcutaneous spaces. On 1, 2, 3, and 4 weeks, subcutaneous regions of interest were collected and subsequently fixed in 4% paraformaldehyde solution. Each group contained three biologically independent samples. The subcutaneous regions of interest were conducted for histological analyses using hematoxylin and eosin (H&E) staining. H&E staining images were acquired using a digital slide scanner (HAMAMATSU PHOTONICS, NanoZoomer®S360). A certified histopathologist assessed the degree of inflammation under blinded conditions. Blood was also collected for blood routine and blood chemistry. Each group contained three independent rats.

Additionally, the biocompatibility and the effectiveness in wound healing were also evaluated using a rat liver bleeding model. Surgicel® Fibrillar and the FgC6 patch were shaped into disks, with the size of 10 mm in diameter and 3 mm in thickness. An incision injury, measuring 8 mm in length and 3 mm in depth, was created. Surgicel® Fibrillar and the FgC6 patch were applied with manual compression until haemostasis. The abdomen was closed after haemostasis was achieved. At 1, 2, 4 weeks post-implantation, the rats were euthanized. The tissues were collected and fixed in 4% paraformaldehyde solution for subsequent analysis. The histological analyses were conducted using hematoxylin and eosin (H&E) staining, and immunofluorescence analyses performed using CD68/CD206 makers. H&E staining images were acquired using a digital slide scanner (HAMAMATSU PHOTONICS, NanoZoomer®S360). Immunofluorescence images were acquired using a digital microscopy (3DHistech, PANNORAMIC MIDI). Rabbit multiclonal antibody to anti-CD 68 (Abcam, ab303565, 1:400 for Immunofluorescence); Rabbit monoclonal antibody to anti-CD 206 (Cell Signaling Technology, #24595, 1:200 for Immunofluorescence); Alexa Fluor 488 labeled anti-rabbit secondary antibody (Life technologies, A21206, 1:400 for Immunofluorescence). The fluorescence intensity of the expressed antibodies was measured and quantified via ImageJ (version 2.9.0). Each group contained three biologically independent replicates. Blood was also collected for blood routine and blood biochemistry. Each group contained three independent rats.

## In vivo haemostatic performance on porcine liver injury model

The haemostatic performance was evaluated using porcine liver injury model. This study was approved by the Laboratory Animal Welfare and Ethics Committee of The Second Affiliated Hospital of Zhejiang University School of Medicine (SAHZU2024#248). Female Bama pigs (22-26 kg, Shanghai Jiagan Biotechnology Co., Ltd) were utilized for porcine liver injury model. General anesthesia was administered for all animal procedures, with maintenance of anesthesia using propofol (0.1-0.2 mg/kg/min). The weighed medical gauze was placed beneath the liver. A part of the liver lobe (ca. 50 mm in length) was cut off to establish the porcine liver injury model (two independent injuries per pig; ten pigs). And free bleeding was allowed for 30 s (pretreatment blood loss) to assess bleeding rates. Then the pretreatment blood was wiped by weighed medical gauze before the haemostatic materials were used on the wound. FgC6 patch was directly applied on the wound with manual compression for 30 s. Commercial products (Surgicel® Fibrillar, Surgiflo®(Thrombin), TachoSil®) were used according to manufacturer's guidelines (Supplementary Table 7). The compression was interrupted after the certain time intervals to check for the haemostasis (Supplementary Table 8). Each groups contained five independent injuries. When blood did not spread on the medical gauze or around the wound, it was indicative of successful haemostasis. The time was recorded immediately. These haemostatic materials were collected, and absorbed blood on weighed medical gauze or haemostatic materials was recorded as posttreatment blood loss. The pigs were monitored for 3 h, and finally euthanized with a lethal dose of potassium chloride solution under general anesthesia.

## In vivo haemostatic performance on porcine femoral artery injury model

The haemostatic performance was evaluated using porcine femoral artery injury model. All animals were treated according to the standard guidelines approved by the Laboratory Animal Welfare and Ethics Committee of The Second Affiliated Hospital of Zhejiang University School of Medicine (SAHZU2024#248). Female Bama pigs (22-26 kg, Shanghai Jiagan Biotechnology Co., Ltd) were utilized for porcine femoral artery injury model. General anesthesia was administered for all animal procedures, with maintenance of anesthesia using propofol

(0.1-0.2 mg/kg/min). A severe arterial haemorrhage was produced with 6 mm arteriotomy (one injury per pig; twelve pigs), and unrestricted bleeding was allowed for 45 s (pretreatment blood loss) to assess bleeding rates. Then the pretreatment blood was wiped by weighed medical gauze from the inguinal cavity before the haemostatic materials were used on the wound. FgC6 patch was directly applied on the wound with manual compression for 1 min. Commercial products (Combat Gauze®) was used according to manufacturer's guidelines (Supplementary Table 7). The compression was interrupted after the certain time intervals to check for the haemostasis (Supplementary Table 9). Each group contain six independent injuries (pigs). When blood did not spread on the medical gauze or around the wound, it was indicative of successful haemostasis. The time was recorded immediately. These haemostatic materials were collected, and absorbed blood on weighed medical gauze or haemostatic materials was recorded as posttreatment blood loss. The survival of injured pigs was observed within 3 h. After being observed for 3 h, the pigs were finally euthanized with a lethal dose of potassium chloride solution under general anesthesia.

## Statistical analysis

Data were presented as mean ± SD. In the statistical analysis for comparison between multiple samples, a one-way ANOVA followed by Tukey's post-hoc test was conducted. In the statistical analysis between two data groups, a two-sided Student's t-test was used. Statistical analysis was performed using OriginLab (version 2019b). The probability value ($P$) < 0.05 was considered to indicate significant difference.

## Reporting summary

Further information on research design is available in the Nature Portfolio Reporting Summary linked to this article.

## Data availability

The authors declare that all data needed to support the findings of this study are provided within the article, Supplementary information, and Source data file. This study utilizes publicly accessible data from the Protein Data Bank (PDB) under accession code: 3GHG. Source data is available for Figs. 1–6, and Supplementary Figs. 3–7. Source data are provided with this paper.

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

## Acknowledgements

We gratefully acknowledge the financial support by National Natural Science Foundation of China (No. 82202483, L.Y.; 82272860, Y.D.; 52425305, Z.M.; U23A202181, W.W.), Key Research and Development Program of Zhejiang Province (No. 2024C03143, Y.D.), Central Guidance on Local Science and Technology Development Fund of Zhejiang Province (2023ZY1017, W.W.), Key Project of Traditional Chinese Medicine Science and Technology Plan of Zhejiang Province (GZY-ZJ-KJ-24077, Y.D.). We thank the staff members of BL19U2 beamline (https://cstr.cn/31129.02.NFPS.BL19U2) at the National Facility for Protein Science in Shanghai (https://cstr.cn/31129.02.NFPS), for providing technical support and assistance in data collection and analysis. We acknowledge the Testing and Analysis Center of Department of Polymer Science and Engineering at Zhejiang University for technical support. We thank Dr. Xiaoqiang Jin and Feng Cai for their assistance in comparing the commercial products with FgC6 patch.

## Author contributions

Z.M., L.Y., and Z.L. conceived the idea and designed the study. L.Y. synthesized and characterized the protein patch. Z.L. carried out the in vitro cytotoxicity experiments. Y. Z conducted the SAXS analysis. L.Y., Z.L., Z.T., and Y.H.D. performed the in vivo studies. Z.M., Y.D. and W.W. supervised and directed the project. All authors analyzed and interpreted the results. L.Y., Z.L. and Z.M. and wrote the manuscript with inputs from all authors.

## Competing interests

Zhejiang University has filed for patent protection on the materials and methods described herein, and Z.M., L.Y., and Y.D. are named as inventors (application number: 202310515045.1). The remaining authors declare no competing interests.
