## [Transparent Peer Review file · Nature Communications]

Molecular self-assembly strategy tuning a dry crosslinking protein patch for biocompatible and biodegradable haemostatic sealing

Corresponding Author: Professor Zhengwei Mao

Version 0:

Reviewer comments:

Reviewer #1

(Remarks to the Author)

This manuscript describes a new hemostatic sponge patch based on modifications to fibrin sponge patches. It is noteworthy that the material has strong hemostatic ability. Overall, the novelty is low, as there are numerous biomaterials in development and marketed that are similar.

Modifications to fibrin and fibrin sponge patches are common throughout the literature for hemostatic applications. While the specific method of modification of fibrin presented here may be new, I categorize this as a moderate extension of current literature rather than a novel material or method.

While the FgC6 sponge patch stopped bleeding better than controls, the controls are not best-in-class. Best-in-class product would be EVARREST™ Fibrin Sealant Patch, which would be a direct competitor and widely used operating rooms. Their experiment didn't do any direct comparison between their product and EVARREST patch. Floseal would also be useful best-in-class for liver injury. The manuscript compared FgC6 to Surgiflo, but that did not contain thrombin like best-in-class Floseal would contain.

The comparison to Combat Gauze in the femoral artery bleed is problematic. Combat Gauze is the gold-standard in swine femoral artery models, and usually works with near 100% success to maintain survival for 3 hr, which is described extensively in the literature. The failure of Combat Gauze demonstrates that this model was not done to normal standards and was likely biased to favor FgC6.

Reviewer #2

(Remarks to the Author)

In this article, the authors characterize a fibrinogen based patch to understand its hemostatic efficacy and adhesion performance on wet tissue. They show efficacy of the patch in multiple in vivo models including a liver injury, abdominal aorta and femoral artery bleeding injuries. However, several items need to be addressed prior to publication.

- The introduction section does not provide sufficient background information about the material used for the patch itself. How fibrinogen is modified needs to be more clearly articulated.
- Section 2.1 should describe which hydrophobic groups were grafted onto lysine residues of Fg.
- Line 130: The authors note that FgC6 required high concentrations to ensure protein entanglement and inter-molecular interactions, typically 50 mg/mL or above. These concentrations are similar to those required for fast gelation of fibrin glue however such high concentrations of fibrinogen result in a dense fibrin structure that can impede healing. The authors should comment on the use of such a high concentration of fibrinogen in their system.
- Figure 1 shows thrombin-catalyzed release for FgC6 compared to Fg but it would be useful to show additional in vitro clotting parameters here.

- Line 272: The authors should comment on the source of fibrinogen used for their patch.
- It is unclear if statistics were run for the data shown in Figure 3. If so it should be clearly indicated on the graphs.
- Lines 408-425: The authors compare FgC6 to several commercially available hemostatic agents/adhesives. This section is written as though they directly compared FgC6 to these various adhesives (referred to as reference groups) but Supplemental Table 4 simply summarizes the characteristics of these various products.
- Lines 575-578: Magnification used for TEM should be specified.
- In the materials and methods section, in general the number of replicates for each assay should be reported.
- Line 655: The company that manufactured the FDP ELISA should be noted.
- Line 669: The histological stain(s) used for tissue analysis should be noted.
- Line 718: How hemostasis was defined should be noted.

Reviewer #3

(Remarks to the Author)

The authors report a modified fibrinogen (FgC6) based hemostatic patch for bleeding control. An extensive set of morphological, mechanical, and biological characterizations was provided to evaluate the FgC6 hemostatic patch. Pre-clinical study models are appropriate with data and analyses sufficiently provided. All data and analyses expected for hemostatic biomaterials were thoroughly provided in the current manuscript. It's a rare occasion that the reviewer does not have comments and support the publication of the manuscript without further revision.

Version 1:

Reviewer comments:

Reviewer #1

(Remarks to the Author)

This manuscript develops a new hemostatic sponge patch based on modifications to fibrin sponge patches. It is noteworthy that the material rapidly stops small bleeds. Overall, the novelty is low, as there are numerous biomaterials in development and marketed that are similar. The revised manuscript does not change my opinion on the novelty and suitability for Nature Communications. The level of novelty is more appropriate for a specialized journal. Furthermore, while many models are presented in the manuscript, they were performed in non-standard ways that bias the FgC6.

1) Modifications to fibrin and fibrin sponge patches are common throughout the literature for hemostatic applications. While the specific method of modification of fibrin presented here may be new, I categorize this as extension of current literature rather than a novel material or method. In the revision there are claims that because the FgC6 material does not incorporate thrombin then it is novel, and that thrombin adds complexity. There are other non-thrombin fibrinogen materials marketed for surgical use. The claim that thrombin adds complexity is not convincing to me, as there are many thrombin products that have been approved by FDA, and are used extensively during surgical procedures effectively and safely (they do not regularly cause thrombosis as claimed in the revision). In my opinion, it is not convincing that a somewhat easier manufacturing procedure is particularly novel.

2) While many animal models were used to compare FgC6 to other materials, such as EVARREST™ Fibrin Sealant Patch, fibrin glue, and Comat Gauze the experiments were not performed to reasonable standards. All of the bleeds tested are small bleeds. The bleeding models are not normal, as most bleeding models have larger amounts of blood loss. The models do not mimic surgical bleeds nor traumatic hemorrhage. For example, Combat Gauze is the gold-standard in swine femoral artery models, and usually works with 100 % success in extensive literature publications. The failure of Combat Gauze demonstrates that the pig model was not done correctly and was likely biased to favor FgC6. The model was done incorrectly because only a small 1 mm puncture was made and bleeding was severe enough (or assessed long enough) to assess survival or real hemorrhage. The correct and widely-used model to mimic surgical bleeds or traumatic hemorrhage is a bleed that results in death of the pig. In the rabbit model, the blood loss was less than a 1 mL, which is not relevant to the types of bleeding that need novel materials. Overall, I have the impression that the experiments were biased to show a positive difference by FgC6, rather than testing if FgC6 is particularly effective at stopping clinically relevant bleeds better than current products.

3) The models were also done without following the instructions of the packaging of the hemostatic controls. They did not compress the materials onto the tissues for the required time, for example. In Figures 6 and 13, the competitor dressings were not applied correctly—Combat Gauze was only pressed down for 30 seconds, and Everast for just 10 seconds. While I understand the authors' goal of highlighting their product's ability to stop bleeding within 30 seconds and its lack of long-term effects, the data does not reflect a fair comparison when the competitors' products are not used as intended. Hemostatic treatment typically involves more than just the initial 30 seconds of application.

4) The manuscripts noted that Combat Gauze is not considered a hospital standard, however it is used extensively for abdominal packing and other surgical applications.

Reviewer #2

(Remarks to the Author)

The authors have responded to my previous comments and I have no further questions.

Version 2:

Reviewer comments:

Reviewer #1

(Remarks to the Author)

My concerns have been sufficiently addressed.

Response to the Comments from the Reviewers:

We appreciate the reviewers for positive comments and the feedbacks provided to strengthen the manuscript. We provide a point-by-point response to each of the reviewers. The corresponding changes to the revised Manuscript and Supporting Information have been highlighted in red.

Reviewer #1:

This manuscript describes a new hemostatic sponge patch based on modifications to fibrin sponge patches. It is noteworthy that the material has strong hemostatic ability. Overall, the novelty is low, as there are numerous biomaterials in development and marketed that are similar.

Modifications to fibrin and fibrin sponge patches are common throughout the literature for hemostatic applications. While the specific method of modification of fibrin presented here may be new, I categorize this as a moderate extension of current literature rather than a novel material or method.

Response: We appreciate the reviewer's comments of our work. In our work, a fibrinogen patch was prepared via a molecular self-assembly strategy, in which fibrinogen was modified with hydrophobic groups to induce self-assembly hydrogel, and then converted into a fibrinogen-based patch. We'd like to highlight the significance of our work besides the moderate extension of current literature to address Reviewer#1's concern on the novelty of our work.

Indeed, as Reviewer#1 mentioned, modifications to fibrin and fibrin sponge patches are common throughout the literature for hemostatic applications. The fibrin (glue) is made from a combination of fibrinogen, thrombin, and calcium, in which thrombin converts fibrinogen to fibrin to form glue, and mimics the end reaction of the natural clotting cascade. Its inherently weak adhesion and mechanical performance restrict the effectiveness in hemostatic applications. Besides, the usage of thrombin results in the complexities of preparation and manipulation, and safety concerns. In contrast, our work has developed hydrophobic group modified fibrinogen patch to enhance the mechanical performance and adhesion properties. To date, we have not found a fibrinogen-only patch for hemostatic sealing in the previous studies. Firstly, we summarize the modification

methods to *fibrin and fibrin (glue/sealant)* patches to enhance the adhesion property and mechanical performance in previous literature.

1) Modifications to fibrin (glue/sealant). The fibrin (glue/sealant) is combined with natural polymers (e.g. collagen, alginate, hyaluronic acid, laminin, elastin, and agarose) or synthetic polymers (e.g. polyethylene glycol, polyvinyl alcohol) to regulate the mechanical properties of the fibrin glue and promote wound healing¹. Besides, in natural system, the post-translational modification of fibrinogen (e.g. oxidation, nitration, glycosylation, glycation, acetylation, phosphorylation) has effects on the fibrin glue formation and fibrin clot structure². However, short shelf-life and a need to be mixed before each use of solution state, and time-consuming dissolution of lyophilized fibrinogen/thrombin, limited fibrin glue's usefulness in emergency settings³. Besides, the adhesion property of modified fibrin (glue/sealant) has not been significantly improved.

2) Modifications to fibrin (glue/sealant) patch. The freeze-dried fibrinogen/thrombin are immobilized on carriers (e.g. collagen, oxidized cellulose, polyglactin gauze) to prepare the fibrin (glue/sealant) patches to increase the concentration of coagulation factors around the wound site, which have been manufactured to commercial products (EVARREST[®], TachoSil[®])⁴. The relatively high concentration of coagulation factors can improve the mechanical performance of blood clot (fibrin glue). However, the design of patches emphasizes the necessity of keep dry until use, as moisture might trigger an enzymatic reaction between fibrinogen and thrombin. Additionally, a relatively long period of manual compression (at least 3 min) is required to achieve a high local concentration of coagulation factors and facilitate the formation of fibrin (glue/sealant) to control bleeding⁵. Zhao's research has reported that TachoSil[®] failed to achieve hemostatic sealing bleeding liver in systemic heparin administration, following the manufacturer's guidelines. The relatively poor adhesion performance led to the weak seal between the TachoSil[®] and the surface of the tissue⁶.

The aforementioned modification methods for fibrin and fibrin (glue/sealant) patches, such as incorporating polymer hydrogel networks and carrier scaffolds, can adjust the fibrin (glue/sealant)'s network to enhance their mechanical performance. However, these methods have limited impact on improving the adhesion property. Besides, the usage of thrombin introduces complexities in preparation and manipulation, along with associated safety concerns (e.g. thrombogenic risks).

In our work, a fibrinogen patch was prepared via a molecular self-assembly strategy, in which fibrinogen was modified with hydrophobic groups to induce self-assembly hydrogel, and then converted into a dry cross-linking patch. The fibrinogen-based patch can absorb the interfacial water on the wet tissue, directly transforming into a hydrogel that establishes a tight contact surface with wet tissue. Simultaneously, the strong intra/inter-molecular interaction of modified fibrinogen molecules facilitate robust adhesion to the tissue surface. Compared to the existing modified fibrin (glue/sealant) patch, the simple composition eliminates the complexities in preparation and manipulation, and the associated safety concerns (e.g. thrombogenic risks).

We have highlighted these points to underscore the novelty of this work in Paragraph 2 of the Introduction section in the revised manuscript (Pages 3-4).

“Several protein-based patches have been developed to remove body fluid and improve adhesion performance, such as fibrin sealant patch (EVARREST[®], TachoSil[®])²⁷. However, their intricate manufacturing process, the requirement for prolong application of consistent pressure (at least 3 min) to achieve adhesion, and composition-related safety concerns (e.g. thrombogenic risks) substantially limit their suitability for clinical use”

1. While the FgC6 sponge patch stopped bleeding better than controls, the controls are not best-in-class. Best-in-class product would be EVARREST[™] Fibrin Sealant Patch, which would be a direct competitor and widely used operating rooms. Their experiment didn't do any direct comparison between their product and EVARREST patch. Floseal would also be useful best-in-class for liver injury. The manuscript compared FgC6 to Surgiflo, but that did not contain thrombin like best-in-class Floseal would contain.

Response: We appreciate the reviewer's constructive comments, which have greatly enhanced the quality of our manuscript. Best-in-class products have been considered as referential groups. EVARREST[®] fibrin sealant patch consists of fibrinogen and thrombin powder embedded in oxidized cellulose gauze. Another Commercial fibrin sealant patch, TachoSil[®] consists of fibrinogen and thrombin coated onto a collagen sponge. Floseal[®] is currently unavailable in our country. Floseal[®] and Surgiflo[®] hemostatic share similar components, consisting of gelatin granules and optional thrombin. Surgiflo[®] hemostatic matrix with thrombin serves as a substitute for Floseal[®] as a referential group. In revised manuscript, FgC6 patch has been compared with the best-in-class products including

fibrin sealant patch (EVARREST[®], TachoSil[®]) and hemostatic matrix with thrombin (Surgiflo[®](thrombin)) in adhesion property and hemostatic performance.

1) Adhesion property.

According to the product package inserts, TachoSil[®] and EVARREST[®] fibrin sealant patches have only one active side intended for application to the bleeding surface. The assessment methods used to evaluate the adhesion performance of double-sided adhesive in our study (previous manuscript) may not be suitable for TachoSil[®] and EVARREST[®]. To facilitate a more appropriate comparison between the best-in-class products with FgC6 patch, adhesion assessment methods based on modified version of the American Society for Testing Material (ASTM) F2258-05 and ASTM F2255-05 standards were designed and performed^{7,8}. The non-active side of the fibrin sealant patches (TachoSil[®] and EVARREST[®]) was affixed on a stiff fixture/backing to assess the adhesion property of the active side of the patch. And, any side of FgC6 patch was also affixed on a stiff fixture/backing to ensure a consistent comparison. The adhesion properties to blood and blood covered porcine skin have been evaluated.

FgC6 patch exhibited excellent attachment upon contact with blood (tensile strength of 57.4 ± 8.4 kPa), and displayed robust adhesion (shear strength of more than 31.0 ± 1.6 kPa) to blood-covered porcine skin. As shown in Fig. 3e-3j, FgC6 patch demonstrated superior adhesion performance compared to existing commercially available tissue adhesives, including fibrin sealant patch (EVARREST[®], TachoSil[®]), gelatin-based hemostatic matrix (Surgiflo[®](Thrombin)). These commercially available tissue adhesives exhibited low attachment upon contact with blood (tensile strength less than 8 kPa), and weak adhesion (shear strength less than 10 kPa) to blood-covered porcine skin. FgC6 patch offered excellent adhesion capacity compared to commercially best-in-class products.

The following discussion has been added in the revised manuscript (Page 10), and the data have been updated in Fig. 3e-3j. The previous Fig. 3e-3j have been moved to the revised Supplementary Information (Supplementary Fig. 7).

“The adhesion properties of FgC6 patch were assessed for tensile strength and shear strength based on the modified American Society for Testing Material (ASTM) standards^{11,34}. FgC6 patch exhibited excellent attachment upon contact with blood (tensile strength of 57.4 ± 8.4 kPa), and robust adhesion (shear strength of 31.0 ± 1.6 kPa) to

blood-covered porcine skin, which were considerably higher than those of Fg (tensile strength of 6.2 ± 0.6 kPa and shear strength of 3.3 ± 0.1 kPa, Fig. 3e-3j), respectively. Meanwhile, in comparison to existing available tissue adhesives, the FgC6 patch showed superior adhesion capabilities. It significantly surpassed the performance of fibrin-based sealants (Fibrin Glue), fibrin-based patches (EVARREST[®], TachoSil[®]), gelatin-based hemostatic matrixes (Surgiflo[®](Thrombin)). These commercially available tissue adhesives exhibited low attachment upon contact with blood (tensile strength less than 8 kPa), and weak adhesion (shear strength less than 10 kPa) to blood-covered porcine skin. Furthermore, FgC6 patch offered excellent adhesion capacity compared to other commercially available hemostatic materials (oxidized cellulose based-dressing, Surgicel[®] Fibrillar; gelatin-based dressing, Gelatin Sponge; Supplementary Fig. 7)”

Figure 3. Mechanical property and adhesion performance of FgC6 patch. (e) Schematic of tensile test used for determining tensile strength based on the modified ASTM F2258-05 standard. (f) Force–displacement curves of the tensile test. (g) Tensile strength of Fibrin Glue, Surgiflo[®](Thrombin), Surgicel[®] Fibrillar, EVARREST[®], TachoSil[®], Fg and FgC6, respectively (n = 3). (h) Schematic illustrating the lap shear test used for assessing the shear strength on one side of porcine skin based on the modified ASTM F2255-05 standard. (i) Force–displacement curves of the lap shear test. (j) Shear strength of Fibrin Glue, Surgiflo[®](Thrombin), Surgicel[®] Fibrillar, EVARREST[®], TachoSil[®], Fg and FgC6, respectively on one side of porcine skin (n = 3). P values are

determined by one-way ANOVA followed by the Tukey's comparison test. Error bars, mean \pm SD. ***P < 0.001.

Supplementary Figure 7. Adhesion performance of FgC6 patch. (a) Schematic of determining shear strength on the porcine skin based on standard lap-shear test (ASTM F2255-05). (b) Force–displacement curves of the lap shear test. (c) Shear strength of Fibrin Glue, Surgicel® Fibrillar, Gelatin Sponge, Fg and FgC6, respectively. (d) Schematic of interfacial toughness on the porcine skin based on the standard T-peel test (ASTM F2256-05). (e) Force–displacement curves of the T-peel adhesion test. (f) Interfacial toughness of Fibrin Glue, Surgicel® Fibrillar, Gelatin Sponge, Fg and FgC6, respectively. P values are determined by one-way ANOVA followed by the Tukey's comparison test. Error bars, mean \pm SD, n = 3. ***P < 0.001.

2) Hemostatic performance.

The hemostatic efficacy of FgC6 patch has been compared with that of the fibrin sealant patches (EVARREST®, TachoSil®) and gelatin matrix with thrombin (Surgiflo®(Thrombin)) in a rabbit liver injury penetration model. The samples were applied to the liver incision with manual compression for 10 s. When blood did not spread on the filter paper or around the wound, it was deemed indicative of hemostasis. According to the manufacturer's guidelines, fibrin sealant patches (EVARREST®, TachoSil®) are recommended to applied to a bleeding injury for 3 min. In this study, after manual compression for 10 s, the fibrin sealant patches (EVARREST®, TachoSil®) were

unable to seal the liver incision injury, resulting in blood oozing from the unsuccessful seal between the patches and the liver surface (Fig. 6b). In contrast, FgC6 patch exhibited rapid and strong adhesion to injured site and achieved hemostatic sealing (Fig. 6b,c). Blood loss in the FgC6 patch group (30 ± 8 mg) was about 86% less than that in fibrin sealant patches (EVARREST[®], TachoSil[®]) groups, and 93% less than that in Surgiflo[®](Thrombin). Moreover, FgC6 patch demonstrated significantly higher resistance to detachment, leading to deformation of the tissue surface during the peeling process (Fig. 6c).

The results have been shown in Fig. 6, and the following discussion has been added in the revised manuscript (Page 18).

“The liver incision injuries were sealed by FgC6 patch and commercially available products. The samples were applied to the incision with manual compression for 10 s. When blood did not spread on the filter paper or around the wound, it was deemed indicative of hemostasis. According to the manufacturer’s guidelines, fibrin sealant patches (EVARREST[®], TachoSil[®]) are recommended to applied to a bleeding wound for a duration of 3 min. In this study, after manual compression for 10 s, the fibrin sealant patches (EVARREST[®], TachoSil[®]) were unable to seal the liver incision injury, resulting in blood oozing from the unsuccessful seal between the patches and the bleeding wound (Fig. 6b and Supplementary Fig. 8). In contrast, FgC6 patch exhibited rapid and strong adhesion to the site of injury, effectively achieving hemostatic sealing (Fig. 6b,c). The FgC6 patch group experienced a blood loss of 30 ± 8 mg, which was approximately 86% less than the blood loss in fibrin sealant patches (EVARREST[®], TachoSil[®]) groups, and 93% less than the blood loss in Surgiflo[®](Thrombin). The FgC6 patch also demonstrated superior hemostatic performance in terms of robust adhesion compared to other existing commercially available hemostatic materials (Supplementary Fig. 9 and Supplementary Table 4), including kaolin-based material (Combat Gauze[®]), chitosan-based material (ChitoSAM[®]), oxidized cellulose-based material (Surgicel[®] Fibrillar), gelatin-based material (Gelatin Sponge) and fibrin-based sealant (Fibrin Glue). These commercially available hemostatic materials exhibited limited adhesion performance and low hemostatic efficiency on blood-covered injury (Supplementary Fig. 9). Moreover, FgC6 patch demonstrated significantly higher resistance to detachment, leading to deformation of the tissue surface during the peeling process (Fig. 6c).”

Figure 6. *In vivo* hemostatic sealing performance in a rabbit liver bleeding model. (a) Schematic of the rabbit liver incision bleeding model. An incision of 10 mm in length and 5 mm in depth was made to establish the rabbit liver incision bleeding model. **(b)** Photographs of the hemostatic treatment for liver injuries in the blank control, EVARREST® and FgC6 patch groups, respectively. **(c)** Hemostatic time and **(d)** blood loss of blank control, Surgiflo®(Thrombin), EVARREST®, TachoSil® and FgC6 patch groups. P values are determined by one-way ANOVA followed by the Tukey's comparison test. Error bars, mean ± SD, n = 4. ***P < 0.001.

2. The comparison to Combat Gauze in the femoral artery bleed is problematic. Combat Gauze is the gold-standard in swine femoral artery models, and usually works with near 100% success to maintain survival for 3 hr, which is described extensively in the literature. The failure of Combat Gauze demonstrates that this model was not done to normal standards and was likely biased to favor FgC6.

Response: We are grateful for the reviewer's constructive comments. Combat Gauze®, a kaolin impregnated gauze, is an inorganic mineral that activates Factor XII, triggering and accelerating the body's natural clotting cascade to form a blood clot and control

bleeding. This process of blood clotting needs several minutes. Combat Gauze® is not biodegradable, and is recommended as a pre-hospital hemostatic material to improve the survival rate. In Tactical Combat Casualty Care (TCCC) Guidelines, the applications of hemostatic dressing including Combat Gauze® has been recommended. For massive hemorrhage, Combat Gauze® should be applied with at least 3 min of direct pressure to maintain survival for post-hospital rescue. In our previous study, Combat Gauze® was applied to the injury site with manual compression for 5 min to manage the fatal emergency hemorrhage and enhance survival rate in a swine gunshot model⁹ (Lisha Yu, Hongliang Zhang, Liping Xiao, Jie Fan, Tanshi Li. ACS Appl. Mater. Interfaces 2022, 14, 21814–21821). Combat Gauze® is a highly effective hemostatic material for pre-hospital use, but is not recommended for operating room settings.

Femoral artery bleeding injury is an acute hemorrhage that requires a short period of time to control, otherwise it can be life-threatening. High-pressure and high-volume bleeding from arterial injuries require strong adhesion or effective procoagulant performance to effectively seal the injury. Given the limited availability of hemostatic material for severe hemorrhage, Combat Gauze®, a procoagulant material, was utilized as a referential group for managing massive hemorrhage in porcine femoral artery injury. In this study, we applied the hemostatic materials to the injured femoral artery and performed manual compression for only 30 s to demonstrate the rapid and robust hemostatic sealing of FgC6 patches. Indeed, Combat Gauze® was not applied as the normal standards to maintain the survival (e.g. at least 3 min of manual pressure). Combat Gauze was not able to form a robust blood clot for sealing femoral artery injury within 30 s of manual compression. In contrast, FgC6 patch can induce strong seal of the artery defect within 30 s, restoring the normal blood stream. Combat Gauze® and FgC6 patch each possess unique advantages in managing severe hemorrhage. For a recognized and clean injured site, FgC6 patch can effectively seal the injured femoral artery in operating rooms. In scenarios of unrecognized hemorrhage and the need to manage multiple sources of bleeding, Combat Gauze® can be used by non-healthcare/healthcare personnel to control bleeding and sustain survival.

The data and figures for comparing the hemostatic performance of FgC6 patch and Combat Gauze® have been moved to the Supplementary Information (Supplementary Fig. 13). The following discussion on the hemostatic performance of FgC6 patch has been

revised, and the potential application scenarios for FgC6 patch have been also indicated (Pages 20-21 of the revised manuscript).

“Femoral artery bleeding injury is an acute hemorrhage that requires a short period of time to control, otherwise it can be life-threatening⁴². High-pressure and high-volume bleeding from arterial injuries require strong adhesion or effective procoagulant performance to effectively seal the injury. An injured defect with 1 mm of diameter was made on the femoral artery to establish the porcine femoral artery bleeding injury (Supplementary Fig. 13 and Supplementary Movie 2). The samples were applied to the injury with manual compression for 30 s. In Tactical Combat Casualty Care (TCCC) Guidelines, for massive hemorrhage, Combat Gauze[®] should be applied with at least 3 min of direct pressure to control bleeding. Thus, Combat Gauze[®] was not able to form a robust blood clot for sealing femoral artery injury within 30 s (Supplementary Fig. 13a). In contrast, FgC6 patch can induce strong seal of the artery defect within 30 s, restoring the normal blood stream immediately (Supplementary Fig. 13a-c). 28 days after the hemostatic application, the femoral artery has normal blood stream without any distal thrombosis (Supplementary Fig. 13d). Furthermore, histological analysis of femoral artery at 16 weeks after repair indicates that the femoral artery was healed (Supplementary Fig. 14). For a recognized and clean injured site, FgC6 patch can effectively seal the injured arteries in operating rooms.”

Supplementary Figure 13. *In vivo* hemostatic sealing in porcine femoral artery

bleeding model. (a) Hemostatic sealing in porcine femoral artery bleeding model (injured defect: 1 mm). Combat Gauze[®] failed to achieve hemostatic sealing of femoral artery injury, but FgC6 patch achieve robust hemostatic sealing. (b) Hemostatic time of Combat Gauze[®] and FgC6 patch in the porcine femoral artery injury model. (c) Injured femoral artery treated with saline rinsing in FgC6 patch group. (d) Injured femoral artery injury at 4 weeks after treatment in FgC6 patch group. Error bars, mean \pm SD, n = 3.

Reviewer #2:

In this article, the authors characterize a fibrinogen based patch to understand its hemostatic efficacy and adhesion performance on wet tissue. They show efficacy of the patch in multiple in vivo models including a liver injury, abdominal aorta and femoral artery bleeding injuries. However, several items need to be addressed prior to publication.

1. The introduction section does not provide sufficient background information about the material used for the patch itself. How fibrinogen is modified needs to be more clearly articulated.

Response: We appreciate the reviewer's comments of our work. Several gelatin-based, collagen-based, fibrin-based and other naturally derived patches have been developed to remove body fluid and improve adhesion performance, their intricate manufacturing process, the requirement for prolonged application of consistent pressure (at least 3 min) to achieve adhesion, and composition-related safety concerns (e.g. thrombogenic risks) substantially limit their suitability for clinical use.

Indeed, modifications to fibrin and fibrin sponge patches are common throughout the literature for hemostatic applications. The fibrin (glue) is made from a combination of fibrinogen, thrombin, and calcium, in which thrombin converts fibrinogen to fibrin to form glue, and mimics the end reaction of the natural clotting cascade. Its inherently weak adhesion and mechanical performance restrict the effectiveness in hemostatic applications. Besides, the usage of thrombin results in the complexities of preparation and manipulation, and safety concerns. In contrast, our work has developed hydrophobic group modified fibrinogen patch to enhance the mechanical performance and adhesion properties. To date, we have not found a fibrinogen-only patch for hemostatic sealing in the previous studies. Firstly, we summarize the modification methods to fibrin and fibrin (glue/sealant) patches to enhance the adhesion property and mechanical performance in previous literature.

1) Modifications to fibrin (glue/sealant). The fibrin (glue/sealant) is combined with natural polymers (e.g. collagen, alginate, hyaluronic acid, laminin, elastin, and agarose) or synthetic polymers (e.g. polyethylene glycol, polyvinyl alcohol) to regulate the mechanical properties of the fibrin glue and promote wound healing¹. Besides, in natural system, the post-translational modification of fibrinogen (e.g. oxidation, nitration,

glycosylation, glycation, acetylation, phosphorylation) has effects on the fibrin glue formation and fibrin clot structure². However, short shelf-life and a need to be mixed before each use of solution state, and time-consuming dissolution of lyophilized fibrinogen/thrombin, limited fibrin glue's usefulness in emergency settings³. Besides, the adhesion property of modified fibrin (glue/sealant) has not been significantly improved.

2) Modifications to fibrin (glue/sealant) patch. The freeze-dried fibrinogen/thrombin are immobilized on carriers (e.g. collagen, oxidized cellulose, polyglactin gauze) to prepare the fibrin (glue/sealant) patches to increase the concentration of coagulation factors around the wound site, which have been manufactured to commercial products (EVARREST[®], TachoSil[®])⁴. The relatively high concentration of coagulation factors can improve the mechanical performance of blood clot (fibrin glue). However, the design of patches emphasizes the necessity of keep dry until use, as moisture might trigger an enzymatic reaction between fibrinogen and thrombin. Additionally, a relatively long period of manual compression (at least 3 min) is required to achieve a high local concentration of coagulation factors and facilitate the formation of fibrin (glue/sealant) to control bleeding⁵. Zhao's research has reported that TachoSil[®] failed to achieve hemostatic sealing bleeding liver in systemic heparin administration, following the manufacturer's guidelines. The relatively poor adhesion performance led to the weak seal between the TachoSil[®] and the surface of the tissue⁶.

In our work, we propose a self-assembly hydrogel strategy that involves regulating the intra/inter-molecular interaction between fibrinogen molecules. This regulation is achieved by modifying the fibrinogen with hydrophobic groups via a N-hydroxysuccinimide ester (NHS ester) reaction. The NHS ester-hydrophobic groups reacted with primary amines on fibrinogen to yield stable amide bonds. The self-assembly protein hydrogel was transformed into a dry protein patch, which relies on a dry cross-linking method to minimize the interfacial water and enables rapid cross-linking to the surface, consequently enhancing adhesive strength. Compared to the existing modified fibrin (glue/sealant) patch, the simple composition eliminates the complexities in preparation and manipulation, and the associated safety concerns (e.g thrombogenic risks).

The background information about the material used for the patch itself has been provided in Paragraph 2 of the Introduction section (Pages 3-4).

“Several protein-based patches have been developed to remove body fluid and improve adhesion performance, such as fibrin sealant patch (EVARREST[®], TachoSil[®])²⁷. However, their intricate manufacturing process, the requirement for prolonged application of consistent pressure (at least 3 min) to achieve adhesion, and composition-related safety concerns (e.g. thrombogenic risks) substantially limit their suitability for clinical use.”

And the modification method of fibrinogen has been clearly articulated in Paragraph 3 of the Introduction section (Page 4).

“This regulation is achieved by modifying the fibrinogen with hydrophobic groups via a N-hydroxysuccinimide ester (NHS ester) reaction.”

2. Section 2.1 should describe which hydrophobic groups were grafted onto lysine residues of Fg.

Response: Thanks for the reviewer's comments. Hexanoyl group (C6) was grafted onto lysine residues of Fg. In the Section 2.1 of the revised manuscript, we have described the modification of the hexanoyl group (C6) on Fg as an illustrative example (Page 5).

“As an illustration, the hexanoyl group (C6) was conjugated to the lysine residues of Fg (denoted as FgC6).”

3. Line 130: The authors note that FgC6 required high concentrations to ensure protein entanglement and inter-molecular interactions, typically 50 mg/mL or above. These concentrations are similar to those required for fast gelation of fibrin glue; however, such high concentrations of fibrinogen result in a dense fibrin structure that can impede healing. The authors should comment on the use of such a high concentration of fibrinogen in their system.

Response: We appreciate the reviewer's comments on our work. In our research, FgC6 hydrogel with 50 mg/mL was used for preparing the dry FgC6 patch. For commercially available fibrin glue, there are various concentrations in the products from 50 to 106 mg/mL (Table X). The concentration of fibrinogen directly influences the mechanical properties of the network. As the reviewer mentioned, a dense fibrin structure could potentially hinder the healing process, raising valid concerns. The fact that most commercially available fibrin glues contain high concentrations of fibrinogen suggests

that these concerns can be either be mitigated or that the tissue-sealing efficacy outweighs the risks associated with such structures.

The biocompatibility of the FgC6 patch were evaluated by in a rat liver injury model. Macrophages play a key role in all stages of wound healing, including inflammation, proliferation, and remodeling. Throughout the healing process, the local macrophage population transitions from predominantly pro-inflammatory (M1 macrophages) to anti-inflammatory (M2 macrophages). M1 macrophages promote immune response in the initial stage of wound repair, and combat foreign pathogens and bacteria. M2 macrophages primarily contribute to tissue repair in the later stage of wound healing. Marker CD68 for pan-macrophage revealed that FgC6 patch and Surgicel® Fibrillar induced comparable levels of inflammation. Immunofluorescence intensity analysis of marker CD206 for M2 type macrophage manifested that FgC6 patch could potentially promote wound healing (Fig. 5e,g). Further in-depth quantitative assessment of FgC6 patch on wound healing needs to be investigated.

The following comment on use of a high centration of fibrinogen in our system has been added in the revised manuscript (Page 16).

“Further in-depth quantitative assessment of FgC6 patch on inflammation regulation and wound healing needs to be evaluated prior to its clinical applications. This includes investigating the dense structure formed by high concentrations of FgC6 and its impact on the wound healing process.”

Table X. Components of fibrin glue/patch currently approval by the FDA, EMA and NMPA³.

Name	Form	Fibrinogen	Thrombin	Factor XIII
Tisseel®	glue	67-106 mg/mL	400-625 IU/mL	0.6-5 IU/mL
Artiss®	glue	67-106 mg/mL	2.5-6.5 IU/mL	0.6-5 IU/mL
Evicel®	glue	55-85 mg/mL	800-1200 IU/mL	9 IU/mL
RAAS®	glue	50 mg/mL	500 IU/mL	None
FgC6 before drying	glue	50 mg/mL	None	None

EVARREST®	patch	8.6 mg/cm ²	37.5 U/cm ²	None
TachoSil®	patch	3.6-7.4 mg/cm ²	1.3-2.7 U/cm ²	None
FgC6	patch	5-16 mg/cm ²	None	None

Figure 5. Biocompatibility and wound healing efficacy of FgC6 patch. (e) Representative immunofluorescence images for Surgicel® Fibrillar at 2 weeks, and FgC6 patch at 2, 4 weeks after hemostatic treatment. (g) Fluorescence intensity measured from the immunofluorescence images of the Surgicel® Fibrillar and FgC6 patch group for CD206 makers (n = 3). P values are determined by a Student's t-test. Error bars, mean ± SD. ***P < 0.001.

4. Figure 1 shows thrombin-catalyzed release for FgC6 compared to Fg but it would be useful to show additional in vitro clotting parameters here.

Response: Thanks for reviewer's helpful suggestion. The additional in vitro clotting parameters have been demonstrated in Supplementary Fig. 3. Fibrin clot formation is initiated by the thrombin-catalyzed release of FpA and FpB. The critical importance of FpA and FpB release contributes to the enzymatic conversion of fibrinogen to fibrin clot. The crosslinking time for fibrin clot formation has been investigated in the Fg and FgC6 system (10 mg/mL), suggesting that FgC6 maintained the capacity to interact with thrombin and the crosslinking time of fibrin clots.

The following text has been modified to improve the rigor of experimental results and discussion in revised version (Page 6).

“FgC6 maintained its capacity to interact with thrombin and its formation of fibrin clots, underscoring the preservation of its bio-functional properties (Fig. 1h and Supplementary Fig. 3).”

Supplementary Figure 3. Crosslinking time of fibrin clot formation in the Fg and FgC6 system (10 mg/mL). Fibrin clot formation is initiated by the thrombin-catalyzed release of FpA and FpB, which contributes to the conversion of fibrinogen to fibrin network. P values are determined by a Student’s t-test. Error bars, mean \pm SD, n = 3; ns: not significant.

5. Line 272: The authors should comment on the source of fibrinogen used for their patch.

Response: Thanks for reviewer's helpful suggestion.

1) The fibrinogen used for patch was sourced commercially and can be obtained from either bovine fibrinogen (Shanghai Yeasen Biotechnology Co., Ltd.) or human fibrinogen (Sigma-Aldrich). Bovine and human fibrinogen are both extracted from the healthy plasma. And both of bovine and human fibrinogen have a molecular weight of about 340 kDa, which are composed of two symmetrical halves, each containing three polypeptide chains α , β and γ . There are only few differences in amino acid sequences and quantities of bovine and human fibrinogen. Despite their different species, bovine and human fibrinogen maintain a high degree of molecular and functional identity, which allows them to serve as substitutes or reference standards for each other in research and clinical applications.

2) In our research, bovine fibrinogen was mainly utilized to prepare the patch. We conducted a comprehensive evaluation of its properties, including a detailed

characterization, mechanical performance, adhesion capability, biocompatibility, biodegradability and hemostatic effectiveness of the patch. Due to the unavailability of complete 3D structural elucidation and FpA/FpB/FDP ELISA assays for bovine fibrinogen, human fibrinogen was utilized for the MD simulation, LC-MS/MS analysis, and FpA/FpB/FDP ELISA assay. The source of fibrinogen used for patch has been clearly stated in the Materials and Methods section (Page 23).

“In this study, bovine fibrinogen was mainly utilized to prepare the FgC6 patch. We conducted a comprehensive evaluation of bovine FgC6 patch, including detailed characterizations, mechanical performance, adhesion capability, biocompatibility, biodegradability and hemostatic effectiveness. Due to the unavailability of complete 3D structural elucidation and FpA/FpB/FDP ELISA assays for bovine fibrinogen, human fibrinogen was utilized for the MD simulation, LC-MS/MS analysis, and FpA/FpB/FDP ELISA assay.”

3) In Line 272 (previous version), the source of fibrinogen used for chemical cross-linking patch, enzymatic cross-linking patch, and FgC6 patch were commercial bovine fibrinogen. Three types of fibrinogen-based patch were prepared to compare their adhesion performance. We have commented the source of fibrinogen used for three patches in revised manuscript to address reviewer’s concerns and hope that it is now clearer (Page 11).

“Three types of fibrinogen-based patches (FgC6, chemical and enzymatic cross-linking patches) were prepared using the same source of fibrinogen to compare their adhesion performance.”

6. It is unclear if statistics were run for the data shown in Figure 3. If so it should be clearly indicated on the graphs.

Response: Thanks for your suggestion. The statistical analysis of Figure 3 has been clearly indicated on the graphs and figure legends. And statistical analysis of other figures and figure legends has been checked and clearly indicated.

Figure 3. Mechanical property and adhesion performance of FgC6 patch. (a) Schematic illustrations for tight contact surface via interfacial water absorption, and robust adhesion to tissue via strong intra/inter-molecular interaction. (b) Water absorption kinetic of Fg and FgC6 systems ($n = 4$). (c) transformation of Fg powder and FgC6 patch systems into the hydrogel or solution state using rheological analysis through dynamic time-sweep tests. (d) Final storage modulus (G') and loss modulus (G'') of Fg solution and FgC6 hydrogel transformed from Fg powder or FgC6 patch, respectively ($n = 3$). (e) Schematic of tensile test used for determining tensile strength based on the modified ASTM F2258-05 standard. (f) Force–displacement curves of the tensile test. (g) Tensile

strength of Fibrin Glue, Surgiflo[®](Thrombin), Surgicel[®] Fibrillar, EVARREST[®], TachoSil[®], Fg and FgC6, respectively (n = 3). (h) Schematic illustrating the lap shear test used for assessing the shear strength on one side of porcine skin based on the modified ASTM F2255-05 standard. (i) Force–displacement curves of the lap shear test. (j) Shear strength of Fibrin Glue, Surgiflo[®](Thrombin), Surgicel[®] Fibrillar, EVARREST[®], TachoSil[®], Fg and FgC6, respectively on one side of porcine skin (n = 3). P values are determined by one-way ANOVA followed by the Tukey's comparison test. Error bars, mean ± SD. ***P < 0.001.

7. Lines 408-425: The authors compare FgC6 to several commercially available hemostatic agents/adhesives. This section is written as though they directly compared FgC6 to these various adhesives (referred to as reference groups) but Supplemental Table 4 simply summarizes the characteristics of these various products.

Response: We appreciate the reviewer's comments of our work. The statement and Supplemental Table 4 comparing between FgC6 to commercially available hemostatic agents/adhesives may be unclear in the previous manuscript. In the revised version, we categorized the commercially available hemostatic materials into five groups based on their composition: fibrin-based materials (EVARREST[®], TachoSil[®], Fibrin Glue), gelatin-based material (Gelatin Sponge, Surgiflo[®](Thrombin)), oxidized cellulose-based material (Surgicel[®] Fibrillar), kaolin-based material (Combat Gauze[®]) and chitosan-based material (ChitoSAM[®]). We further classified the fibrin-based materials according to their form into two types: fibrin sealant (Fibrin Glue) and fibrin sealant patch (EVARREST[®], TachoSil[®]). In the revised version, Supplementary Table 4 summarizes the categories and characteristics of commercial products, and Figure 6 and Supplementary Figure 9 demonstrates the hemostatic performance of commercial products in rabbit liver injury penetration model. The revised manuscript presents the best-in-class hemostatic materials (EVARREST[®], TachoSil[®]) in rabbit liver injury penetration model, as illustrated in Figure 6. Additionally, other hemostatic materials are detailed in Supplementary Material (Supplementary Figure 9).

In the Results and Discussion section of the revised manuscript (Pages 18-19), we have added the following discussion to replace the previous text.

“The liver incision injuries were sealed by FgC6 patch and commercially available products. The samples were applied to the incision with manual compression for 10 s. When blood did not spread on the filter paper or around the wound, it was deemed indicative of hemostasis. According to the manufacturer’s guidelines, fibrin sealant patches (EVARREST[®], TachoSil[®]) are recommended to applied to a bleeding wound for a duration of 3 min. In this study, after manual compression for 10 s, the fibrin sealant patches (EVARREST[®], TachoSil[®]) were unable to seal the liver incision injury, resulting in blood oozing from the unsuccessful seal between the patches and the bleeding wound (Fig. 6b and Supplementary Fig. 8). In contrast, FgC6 patch exhibited rapid and strong adhesion to the site of injury, effectively achieving hemostatic sealing (Fig. 6b,c). The FgC6 patch group experienced a blood loss of 30 ± 8 mg, which was approximately 86% less than the blood loss in fibrin sealant patches (EVARREST[®], TachoSil[®]) groups, and 93% less than the blood loss in Surgiflo[®] (Thrombin). The FgC6 patch also demonstrated superior hemostatic performance in terms of robust adhesion compared to other existing commercially available hemostatic materials (Supplementary Fig. 9 and Supplementary Table 4), including kaolin-based material (Combat Gauze[®]), chitosan-based material (ChitoSAM[®]), oxidized cellulose-based material (Surgicel[®] Fibrillar), gelatin-based material (Gelatin Sponge) and fibrin-based sealant (Fibrin Glue). These commercially available hemostatic materials exhibited limited adhesion performance and low hemostatic efficiency on blood-covered injury (Supplementary Fig. 9). Moreover, FgC6 patch demonstrated significantly higher resistance to detachment, leading to deformation of the tissue surface during the peeling process (Fig. 6c).”

Figure 6. *In vivo* hemostatic sealing performance in a rabbit liver bleeding model. (a) Schematic of the rabbit liver incision bleeding model. An incision of 10 mm in length and 5 mm in depth was made to establish the rabbit liver incision bleeding model. **(b)** Photographs of the hemostatic treatment for liver injuries in the blank control, EVARREST® and FgC6 patch groups, respectively. **(c)** Hemostatic time and **(d)** blood loss of blank control, Surgiflo®(Thrombin), EVARREST®, TachoSil® and FgC6 patch groups. P values are determined by one-way ANOVA followed by the Tukey's comparison test. Error bars, mean ± SD, n = 4. ***P < 0.001.

Supplementary Figure 9. *In vivo* hemostatic sealing in a rabbit liver bleeding model. (a) Schematic of the rabbit liver bleeding model. An incision (length: 10 mm, depth: 5 mm) was made on the liver to establish the rabbit liver bleeding model. (b) Experimental images of the hemostatic treatment of liver injuries in the blank control, Surgicel® Fibrillar, Gelatin Sponge and FgC6 patch groups. (c) Hemostatic time and (d) blood loss of blank control, Combat Gauze®, ChitoSAM®, Surgicel® Fibrillar, Gelatin Sponge, Fibrin Glue and FgC6 patch, respectively. P values are determined by one-way ANOVA followed by the Tukey's comparison test. Error bars, mean \pm SD, n = 4. ***P < 0.001.

Supplementary Table 4. Comparison of commercially available hemostatic materials.

Hemostatic material	Product Name	Major Components	Advantages	Disadvantages
Fibrinogen patch	FgC6 patch (this study)	Fibrinogen patch via protein molecular self-assembly strategy: fibrinogen modified with hydrophobic groups	 — Robust adhesion — Biocompatibility — Biodegradability — No preparation required 	 — Relatively high cost
Fibrin sealant patch	EVARREST®	Fibrinogen-thrombin-embedded oxidized cellulose patch: oxidized regenerated cellulose and polygactin 910 non-woven fibers (carrier), fibrinogen and thrombin (active ingredient)	 — Biocompatibility — Biodegradability — No preparation required 	 — Compression-related side effects — Not use to bleeding from large defects in blood vessels
Fibrin sealant patch	TachoSil®	Fibrinogen-thrombin-impregnated collagen patch: collagen (carrier), fibrinogen and thrombin (active ingredient)	 — Biocompatibility — Biodegradability — No preparation required 	 — Poor fixation to tissue (by limited cross-linking with the collagen carrier)
Fibrin sealant	Fibrin Glue (Tisseel®, Evicel®, Artiss®, RAAS®)	Fibrinogen, thrombin	 — Biocompatibility — Biodegradability — Fast gelation 	 — Not be applied into blood vessels — Low adhesion
Oxidized cellulose dressing	Surgicel® Fibrillar	Oxidized regenerated cellulose	 — Bactericidal property — Biocompatibility 	 — Low hemostasis efficiency — Low adhesion
Gelatin Sponge	Gelfoam®, Gelfilm®, Xiangen®	Gelatin	 — Biodegradability — Low immunogenicity 	 — Compression-related side effects — Low hemostasis efficiency

Gelatin matrix	Surgiflo®	Gelatin matrix	 — Providing tamponade effect — Low immunogenicity 	 — Not be applied into blood vessels — Low adhesion
Gelatin matrix with thrombin	Surgiflo®(Thrombin)	Gelatin matrix, thrombin (active ingredient)	 — Providing tamponade effect — Low immunogenicity 	 — Not be applied into blood vessels — Low adhesion
Gelatin matrix with thrombin	Floseal®	Gelatin matrix, thrombin (active ingredient)	 — Providing tamponade effect — Low immunogenicity 	 — Not be applied into blood vessels — Low adhesion
Chitosan dressing	ChitoSAM®	Non-woven chitosan	 — Various hemostatic mechanisms — Antimicrobial property — Stop severe bleeding 	 — Inadaptation for internal hemostasis — Coagulation-dependent hemostasis
Mineral kaolin dressing	Combat Gauze®	Non-woven gauze (carrier), kaolin (active ingredient)	 — Procoagulant activity — Low cost — Good storability 	 — Inadaptation for internal hemostasis — Coagulation-dependent hemostasis

^a The major components and advantages/disadvantages of hemostatic materials were mainly obtained from their package inserts.

8. Lines 575-578: Magnification used for TEM should be specified.

Response: Thanks for your suggestion. We have specified the magnification used for TEM in Figure 1e (Page 7 in the revised manuscript). Figure 1e and its figure legend have been corrected as follow.

Figure 1. (e) Low magnification (2600 \times) and high magnification (73000 \times) cryogenic transmission electron microscopy (cryo-TEM) images of the Fg and FgC6 systems, respectively. In the FgC6 hydrogel, fibers are observed, formed through protein self-assembly. This contrasts with the homogeneously dispersed protein in the Fg solution.

And following text has been added in Materials and Methods section (Page 24 in the revised manuscript).

“Low magnification (2600 \times) and high magnification (73000 \times) cryo-TEM images were captured for the Fg and FgC6 systems, respectively.”

9. In the materials and methods section, in general the number of replicates for each assay should be reported.

Response: Thanks for your suggestion. The number of replicates for each assay have been reported in the Materials and Methods section.

10. Line 655: The company that manufactured the FDP ELISA should be noted.

Response: Thanks for your suggestion. The company that manufactured the FDP ELISA (Hefei Laier Biotechnology Co., Ltd.) have been noted in the Materials and Methods Section (Page 26 in the revised manuscript).

11. Line 669: The histological stain(s) used for tissue analysis should be noted.

Response: Thanks for your suggestion. Hematoxylin and eosin (H&E) staining for histological analysis has been noted (Page 27 in the revised manuscript).

12. Line 718: How hemostasis was defined should be noted.

Response: Thanks for your suggestion. The hemostasis has been defined in the revised version. The filter paper or medical gauze was place beneath the liver in rabbit liver injury penetration model, pig liver incision injury model and pig liver lobe injury model; and filter paper or medical gauze was place around the wound in rat abdominal aorta injury model and pig femoral artery injury model. When blood did not spread on the filter paper, medical gauze or around the wound, it was indicative of successful hemostasis.

The definition of hemostasis has been noted in rabbit liver injury penetration model and porcine hepatectomy model of the Materials and Methods section (Page 28 in the revised manuscript).

“The filter paper was placed beneath the liver. An incision of 10 mm in length and 5 mm in depth was made on the liver. Subsequently, the samples were applied to the incision site with manual compression for 10 s. When blood did not spread on the filter paper or around the wound, it signified that effective hemostasis had been achieved.”

“The medical gauze was placed beneath the liver. Surgicel® Fibrillar and FgC6 patch were applied on the bleeding sites, respectively. Subsequently, the samples were applied to the incision site with manual compression for 30 s. When blood did not spread on the medical gauze or around the wound, it was indicative of successful hemostasis.”

Reviewer #3:

The authors report a modified fibrinogen (FgC6) based hemostatic patch for bleeding control. An extensive set of morphological, mechanical, and biological characterizations was provided to evaluate the FgC6 hemostatic patch. Pre-clinical study models are appropriate with data and analyses sufficiently provided. All data and analyses expected for hemostatic biomaterials were thoroughly provided in the current manuscript. It's a rare occasion that the reviewer does not have comments and support the publication of the manuscript without further revision.

Response: We appreciate the reviewer's comments of our work. We develop a biocompatible, biodegradable fibrinogen-based patch with robust adhesion achieved via a molecular self-assembly strategy. This patch demonstrates promise as a candidate for clinical hemostatic sealing.

We would like to thank the reviewers again for taking the time to review our manuscript.

Reference

- 1 Sanz-Horta, R. *et al.* Technological advances in fibrin for tissue engineering. *J. Tissue Eng.* **14**, 1-32, (2023).
- 2 de Vries, J. J., Snoek, C. J. M., Rijken, D. C. & de Maat, M. P. M. Effects of post-translational modifications of fibrinogen on clot formation, clot structure, and fibrinolysis. *Arterioscl. Throm. Vas.* **40**, 554-569, (2020).
- 3 Beudert, M., Gutmann, M., Lühmann, T. & Meinel, L. Fibrin sealants: challenges and solutions. *ACS Biomater. Sci. Eng.* **8**, 2220-2231, (2022).
- 4 Spotnitz, W. D. Fibrin sealant: the only approved hemostat, sealant, and adhesive—a laboratory and clinical perspective. *ISRN Surg.*, 203943, (2014).
- 5 Anthis, A. H. C. *et al.* Chemically stable, strongly adhesive sealant patch for intestinal anastomotic leakage prevention. *Adv. Funct. Mater.* **31**, 2007099, (2021).
- 6 Yuk, H. *et al.* Rapid and coagulation-independent haemostatic sealing by a paste inspired by barnacle glue. *Nat. Biomed. Eng.* **5**, 1131-1142, (2021).
- 7 Bao, G. *et al.* Liquid-infused microstructured bioadhesives halt non-compressible hemorrhage. *Nat. Commun.* **13**, 5035, (2022).
- 8 Boerman, M. A. *et al.* Next generation hemostatic materials based on NHS-ester functionalized poly(2-oxazoline)s. *Biomacromolecules* **18**, 2529-2538, (2017).
- 9 Yu, L., Zhang, H., Xiao, L., Fan, J. & Li, T. A bio-inorganic hybrid hemostatic gauze for effective control of fatal emergency hemorrhage in “Platinum Ten Minutes”. *ACS Appl. Mater. Interfaces* **14**, 21814-21821, (2022).

Response to the Comments from the Reviewers:

We appreciate the reviewers for comments and the feedbacks provided to strengthen the manuscript. We provide a point-by-point response to the reviewers. The corresponding changes to the revised Manuscript and Supporting Information have been highlighted in red.

Reviewer #1:

This manuscript develops a new hemostatic sponge patch based on modifications to fibrin sponge patches. It is noteworthy that the material rapidly stops small bleeds. Overall, the novelty is low, as there are numerous biomaterials in development and marketed that are similar. The revised manuscript does not change my opinion on the novelty and suitability for Nature Communications. The level of novelty is more appropriate for a specialized journal. Furthermore, while many models are presented in the manuscript, they were performed in non-standard ways that bias the FgC6.

Response:

Thanks for reviewer's comments. Limited adhesion performance of current fibrin/fibrin sealant patches encourages us to develop novel biomaterials. Although we use the fibrinogen as the main component, the structure, preparation, and mechanism of self-assembly fibrinogen hydrogel in our work are obviously differently from the current fibrin/fibrin patches. The enhanced adhesion performance of FgC6 patch has been confirmed based on the American Society for Testing Material standards. We have conducted a reassessment of the commercial products in strict adherence to the manufacturer's instructions, ensuring a direct and fair comparison with FgC6 patch in in oozing wound (porcine liver resection injury, 50 mm in length) and massive acute hemorrhage (femoral artery bleeding injury, 6 mm incision). The superiority of FgC6 patch have been shown by minimized blood loss and reduced hemostatic time with all products being used as intended.

1) Modifications to fibrin and fibrin sponge patches are common throughout the literature for hemostatic applications. While the specific method of modification of

fibrin presented here may be new, I categorize this as extension of current literature rather than a novel material or method. In the revision there are claims that because the FgC6 material does not incorporate thrombin then it is novel, and that thrombin adds complexity. There are other non-thrombin fibrinogen materials marketed for surgical use. The claim that thrombin adds complexity is not convincing to me, as there are many thrombin products that have been approved by FDA, and are used extensively during surgical procedures effectively and safely (they do not regularly cause thrombosis as claimed in the revision). In my opinion, it is not convincing that a somewhat easier manufacturing procedure is particularly novel.

Response:

Reviewer comments that the modifications to fibrin/fibrin sponge patches are common throughout literatures for hemostatic applications, and there are other non-thrombin fibrinogen materials for surgical use. From these points, Reviewer#1 concludes the a somewhat easier manufacturing procedure is not novel. We disagree the comments, and explain as follow.

(i) We have summarized the modification methods to fibrin and fibrin sponge patches in the last response letter. The current modification methods, such as incorporating polymer hydrogel networks and carrier scaffolds, can adjust the fibrin network to enhance their mechanical performance. However, the enhancement of adhesion property is limited. Several excellent publications in Nature Portfolio have also reported the limited adhesion property of the modified fibrin/fibrin sponge patch products for hemostatic applications (Nature Communications 15, 1215 (2024); Nature Biomedical Engineering, 5, 1131-1142 (2021)). According to package instructions of commercial fibrin sealant patches (EVARREST[®], TachoSil[®]), the requirement for prolong application of consistent pressure (at least 3 min) to achieve adhesion, low efficiency of massive hemorrhage control (e.g. major arterial or venous bleeding), and composition-related safety concerns (e.g. thrombin, thrombogenic risks) substantially limit their versatility for clinical applications.

(ii) The “fibrin” and “fibrinogen” should be distinguished. “Fibrin” is a hydrogel generated from fibrinogen and thrombin. But “fibrinogen”, a protein molecule,

manifests as a powder in its dehydrated state and transitions into a liquid form upon dissolution. Our work proposes a self-assembly hydrogel strategy that involves regulating the intra/inter-molecular interaction between fibrinogen molecules. The self-assembly fibrinogen hydrogel is transformed into a dry protein hydrogel patch, which relies on a dry cross-linking method to minimize the interfacial water and enables rapid cross-linking to the surface, consequently enhancing adhesion strength and mechanical performance. The structure, component, and preparation of self-assembly fibrinogen hydrogel in our work are obviously differently from the current modifications. The adhesion properties of FgC6 patch, Fg (fibrinogen) and best-in-class fibrin/fibrin sponge patch products have been assessed based on the American Society for Testing Material standards (Figure 3), which are standard tests for surgical sealants without any bias (Nature Communications 15, 1215 (2024); Nature Communications 13, 5035 (2022); Nature 575, 169-174 (2019)). FgC6 patch demonstrated superior adhesion performance compared to the Fg group and the best-in-class fibrin/fibrin sponge patch products (EVARREST[®], TachoSil[®], Figure 3).

We have highlighted the above points to manifest the novelty of this work in Paragraph 2 of the Introduction section in the revised manuscript (Page 4).

“Several protein-based patches have been developed to remove body fluid and improve adhesion performance, such as fibrin sealant patch (EVARREST[®], TachoSil[®])²⁷. However, the requirement for prolong application of consistent pressure (at least 3 min) to achieve adhesion, low efficiency of massive hemorrhage control (e.g. major arterial or venous bleeding), and composition-related safety concerns (e.g. thrombin, thrombogenic risks) substantially limit their versatility for clinical applications^{11,28}.”

2) While many animal models were used to compare FgC6 to other materials, such as EVARREST[™] Fibrin Sealant Patch, fibrin glue, and Comat Gauze the experiments were not performed to reasonable standards. All of the bleeds tested are small bleeds. The bleeding models are not normal, as most bleeding models have larger amounts of blood loss. The models do not mimic surgical bleeds nor traumatic hemorrhage. For

example, Combat Gauze is the gold-standard in swine femoral artery models, and usually works with 100 % success in extensive literature publications. The failure of Combat Gauze demonstrates that the pig model was not done correctly and was likely biased to favor FgC6. The model was done incorrectly because only a small 1 mm puncture was made and bleeding was severe enough (or assessed long enough) to assess survival or real hemorrhage. The correct and widely-used model to mimic surgical bleeds or traumatic hemorrhage is a bleed that results in death of the pig. In the rabbit model, the blood loss was less than a 1 mL, which is not relevant to the types of bleeding that need novel materials. Overall, I have the impression that the experiments were biased to show a positive difference by FgC6, rather than testing if FgC6 is particularly effective at stopping clinically relevant bleeds better than current products.

Response:

Thanks for pointing out this question. The animal models have been revised and performed, and the pretreatment blood loss have ensured major bleeding models. A porcine liver resection injury model (50 mm in length, oozing wound) and a porcine femoral artery injury model (6 mm incision, massive acute hemorrhage) were performed to compare hemostatic performance. The protocols of porcine liver resection injury model were based on the reference literatures (*World J Emerg. Surg.* **18**, 19, (2023); *Surgery* **149**, 48-55, (2011)), which were reported for assessing the hemostatic properties of TachoSil® fibrin sealant patch. The protocols of porcine femoral artery injury model were performed according to references (*J Trauma Acute Care Surg.* **68**, 269-278, (2010); *J Trauma Acute Care Surg.* **67**, 450-460, (2009)), which were reported for evaluating the hemostatic performance of Combat Gauze®. These animal models were approved by the Laboratory Animal Welfare and Ethics Committee of The Second Affiliated Hospital of Zhejiang University School of Medicine (SAHZU2024#248).

The free bleeding was recorded before hemostatic treatment. The pretreatment blood loss in oozing wounds (porcine liver resection injury, 50 mm in length) and massive acute hemorrhage (femoral artery bleeding injury, 6 mm incision) were about 20 g and 180 g, respectively (Supplementary Tables 4-5). We have conducted a reassessment of the commercial products in strict adherence to the manufacturer's

instructions, ensuring a direct and fair comparison with FgC6 patch in these two models. The superiority of FgC6 patch have been shown by minimized blood loss and reduced hemostatic time with all products being used as intended. Other inappropriate animal models have been deleted in the revised manuscript.

We have added the following content in the revised manuscript (Pages 17-19), and animal protocols in revised Materials and Methods section (Pages 25-27). The pretreatment blood loss has been recorded in Supplementary Tables 4-5. The hemostatic performance of Surgicel[®] Fibrillar, Surgiflo[®](Thrombin), TachoSil[®] and FgC6 patch in porcine liver injury model has been recorded in Supplementary Videos 1-4, respectively. The hemostatic performance of Combat Gauze[®] and FgC6 patch in femoral artery bleeding injury has been recorded in Supplementary Videos 5-6, respectively.

“2.5 Hemostatic sealing in porcine bleeding models

We demonstrated the rapid hemostatic sealing capabilities of bleeding injuries in pigs, validating the in vivo efficacy of FgC6 patch in a setting that is representative of clinical applications. In the liver resection injury model, a part of the liver lobe, specifically about 50 mm in length was cut off^{41,42}. And free bleeding was allowed for 30 s (pretreatment blood loss) to assess bleeding rates (Supplementary Table 4). Commercially available hemostatic materials, including Surgicel[®] Fibrillar, Surgiflo[®](Thrombin) and TachoSil[®] were applied to bleeding liver injury, following the manufacturer’s guidelines (Supplementary Videos 1-3). Due to strong adhesion performance, FgC6 achieved hemostatic sealing after 30 s of manual compression, showing a significantly reduced hemostatic time and blood loss, compared to Surgicel[®] Fibrillar, Surgiflo[®](Thrombin) and TachoSil[®] (Fig. 6a-c and Supplementary Video 4).

Femoral artery bleeding injury is an acute hemorrhage that requires a short period of time to control, otherwise it can be life-threatening⁴³. High-pressure and high-volume bleeding from arterial injuries require strong adhesion or effective procoagulant performance to effectively seal the injury. An injured defect with 6 mm of incision was made on the femoral artery to establish the porcine femoral artery bleeding injury^{44,45}. And unrestricted bleeding was allowed for 45 s (pretreatment blood loss) to assess

bleeding rates (Supplementary Table 5). Combat Gauze[®], which is recommended for massive hemorrhage in Tactical Combat Casualty Care (TCCC) Guidelines,⁴⁶ was used as a reference group (Supplementary Table 6). Combat Gauze[®] need be applied with at least 3 min of direct pressure to control bleeding. In the porcine femoral artery bleeding injury, it took Combat Gauze[®] over 4 min to effectively achieve hemostasis (Fig. 6d,e and Supplementary Video 5). In contrast, FgC6 patch can induce strong seal of the artery defect within 1 min, restoring the normal blood stream immediately (Fig. 6d, and Supplementary Video 6). Rapid and strong hemostatic sealing obviously reduced the posttreatment blood loss. Thus, the FgC6 patch group experienced a blood loss of 0.4 ± 0.2 g, which was approximately 98% less than the blood loss in Combat Gauze[®] group (Fig. 6e,f).

Therefore, the FgC6 patch can achieve hemostatic sealing in the both oozing wound (such as liver injury) and massive acute hemorrhage (such as femoral artery bleeding injury). This is facilitated by the removal of interfacial water and the strong intra/inter-molecular interaction of FgC6 molecules. Meanwhile, the FgC6 patch exhibited stable sealing ability and excellent biocompatibility.”

Figure 6. *In vivo* hemostatic sealing in porcine injury bleeding model. (a)

Hemostatic treatment in a porcine liver bleeding model. A part of the liver lobe (ca. 50 mm in length) was cut off to establish the porcine liver bleeding model. TachoSil[®] and FgC6 patch were applied on the wound with manual compression until hemostasis achieved. (b) Hemostatic time and (c) blood loss following treatment of Surgicel[®] Fibrillar, Surgiflo[®](Thrombin), TachoSil[®] and FgC6 patch in the porcine liver bleeding model. Each group contains five independent injuries (n = 5). (d) Hemostatic treatment in a porcine femoral artery bleeding model. A severe arterial hemorrhage was produced with 6 mm arteriotomy. Combat Gauze[®] and FgC6 patch were applied on the injured femoral artery with manual compression until hemostasis achieved. (e) Hemostatic time and (f) blood loss following treatment in the porcine femoral artery bleeding model. Each group contains six pigs (n = 6). Data values corresponded to mean ± SD. Error bars represent SD. *P* values are determined by a Student's t-test. ***P* < 0.01, ****P* < 0.001.

“Materials and methods

***In vivo* hemostatic performance on porcine liver injury model**

The hemostatic performance was evaluated using porcine liver injury model. This study was approved by the Laboratory Animal Welfare and Ethics Committee of The Second Affiliated Hospital of Zhejiang University School of Medicine (SAHZU2024#248). Female Bama pigs (22-26 kg, Shanghai Jiagan Biotechnology Co., Ltd) were utilized for porcine liver injury model. General anesthesia was administered for all animal procedures, with maintenance of anesthesia using propofol (0.1-0.2mg/kg/min). The weighed medical gauze was placed beneath the liver. A part of the liver lobe (ca. 50 mm in length) was cut off to establish the porcine liver injury model (two independent injuries per pig; ten pigs). And free bleeding was allowed for 30 s (pretreatment blood loss) to assess bleeding rates. Then the pretreatment blood was wiped by weighed medical gauze before the hemostatic materials were used on the wound. FgC6 patch was directly applied on the wound with manual compression for

30 s. Commercial products (Surgicel[®] Fibrillar, Surgiflo[®](Thrombin), TachoSil[®]) were used according to manufacturer's guidelines (Supplementary Table 7). The compression was interrupted after the certain time intervals to check for the hemostasis (Supplementary Table 8). Each groups contained five independent injuries. When blood did not spread on the medical gauze or around the wound, it was indicative of successful hemostasis. The time was recorded immediately. These hemostatic materials were collected, and absorbed blood on weighed medical gauze or hemostatic materials was recorded as posttreatment blood loss. The pigs were monitored for 3 h, and finally euthanized with a lethal dose of potassium chloride solution under general anesthesia.

***In vivo* hemostatic performance on porcine femoral artery injury model**

The hemostatic performance was evaluated using porcine femoral artery injury model. All animals were treated according to the standard guidelines approved by the Laboratory Animal Welfare and Ethics Committee of The Second Affiliated Hospital of Zhejiang University School of Medicine (SAHZU2024#248). Female Bama pigs (22-26 kg, Shanghai Jiagan Biotechnology Co., Ltd) were utilized for porcine femoral artery injury model. General anesthesia was administered for all animal procedures, with maintenance of anesthesia using propofol (0.1-0.2mg/kg/min). A severe arterial hemorrhage was produced with 6 mm arteriotomy (one injury per pig; twelve pigs), and unrestricted bleeding was allowed for 45 s (pretreatment blood loss) to assess bleeding rates. Then the pretreatment blood was wiped by weighed medical gauze from the inguinal cavity before the hemostatic materials were used on the wound. FgC6 patch was directly applied on the wound with manual compression for 1 min. Commercial products (Combat Gauze[®]) was used according to manufacturer's guidelines (Supplementary Table 7). The compression was interrupted after the certain time intervals to check for the hemostasis (Supplementary Table 9). Each group contain six independent injuries (pigs). When blood did not spread on the medical gauze or around the wound, it was indicative of successful hemostasis. The time was recorded immediately. These hemostatic materials were collected, and absorbed blood on weighed medical gauze or hemostatic materials was recorded as posttreatment blood loss. The survival of injured pigs was observed within 3 h. After being observed for 3

h, the pigs were finally euthanized with a lethal dose of potassium chloride solution under general anesthesia.”

Supplementary Table 4. Bleeding outcomes following injury and treatment in porcine liver injury model

	Pretreatment blood loss (g) ^a	Posttreatment blood loss (g) ^b	Hemostatic time (s)
Surgicel [®] Fibrillar	22.0 ± 4.6	38.5 ± 6.6	258 ± 27
Surgiflo [®] (Thrombin)	22.9 ± 3.0	9.2 ± 3.8	180 ± 4.5
TachoSil [®]	22.2 ± 3.7	5.0 ± 1.3	204 ± 3.3
FgC6	21.5 ± 4.7	1.9 ± 0.9	30 ± 0
P (among groups) ^c	0.957	<0.001	<0.001

^a Free bleeding within 30 s was recorded.

^b Blood loss following treatment was recorded until successful hemostasis.

^c *P* values are determined by one-way ANOVA. Mean ± SD, n = 5.

Supplementary Table 5. Bleeding outcomes following injury and treatment in porcine femoral artery injury model

	Pretreatment blood loss (g) ^a	Posttreatment blood loss (g) ^b	Hemostatic time (s)
Combat Gauze [®]	175.5 ± 39.1	21.0 ± 6.5	330 ± 83
FgC6	188.0 ± 30.9	0.4 ± 0.2	60 ± 0
P (among groups) ^c	0.554	<0.001	<0.001

^a Free bleeding within 45 s was recorded.

^b Blood loss following treatment was recorded until successful hemostasis.

^c *P* values are determined by one-way ANOVA. Mean ± SD, n = 6.

Supplementary Table 7. Methods of application for commercial products from manufacturer’ s guidelines

Product Name	Dosage and administration	Methods of application	Limitations for use/Contraindications	Reference
TachoSil® Fibrin sealant patch	--TachoSil as an adjunct to hemostasis in cardiovascular and hepatic surgery, when control of bleeding by standard surgical techniques (such as suture, ligature or cautery) is ineffective or impractical. --TachoSil is applied to the surface of cardiac, vascular or hepatic tissue only.	(1) Determine the number of patches to be applied by the size of the bleeding area. (2) Select the appropriate TachoSil patch so that it extends 1 to 2 cm beyond the margins of the wound. (3) cleanse the area to be treated. (4) disinfectants and other fluids hold in place with gentle pressure applied through moistened gloves or a moist pad for at least 3 min. (5) To avoid pulling the patch loose, first place a clean surgical instrument at one end of the patch before relieving the pressure	--Not for use in place of sutures or other forms of mechanical ligation in treatment of major arterial or venous bleeding. --Do not apply TachoSil intravascularly. Intravascular application of TachoSil may result in life-threatening thromboembolic events.	Package Insert--- TachoSil [2024]
Surgicel® Fibrillar	--Surgicel Fibrillar is used adjunctively in surgical procedures to assist in the control of capillary, venous and small arterial hemorrhage when ligation or other conventional methods of control are impractical or ineffective.	Use only as much Surgicel Fibrillar as is necessary for hemostasis, holding it firmly in place until bleeding stops.	--Surgicel Fibrillar should not be used to control hemorrhage from large arteries.	Package-Surgicel-[2021]
Surgiflo® (Thrombin)	--Surgiflo, mixed with thrombin solution, is indicated in surgical procedures (other than ophthalmic) as an adjunct to hemostasis when control of bleeding by ligature or other conventional methods is ineffective or impractical.	(1) Identify the source of bleeding. (2) Surgiflo can be used with or without the applicator tip attached to the syringe. Apply sufficient Surgiflo Hemostatic Matrix to cover the entire bleeding surface.	--Do not use Surgiflo in intravascular compartments because of the risk of embolization. --Do not use Surgiflo in closure of skin incisions because it may interfere with the healing of skin edges.	Package-Surgiflo [2020]

		(3) Apply a sterile saline moistened gauze over the Surgiflo to ensure the material remains in contact with the bleeding tissue. (4) After 1-2 min, lift the gauze and inspect the wound site. Once bleeding has ceased, irrigate excess Surgiflo away gently so as not to disturb the new clot.	--Surgiflo should not be used in instances of pumping arterial hemorrhage. It should not be used where blood or other fluids have pooled or in cases where the point of hemorrhage is submerged.	
Combat Gauze®	--Penetrating wounds and massive bleeding --Gunshot wounds --Stabbings --Large non-compressible injuries where tourniquets cannot be applied	(1) Pack Combat Gauze into wound and use it to apply pressure directly over bleeding source (2) Continue to apply pressure for 3 min or until bleeding stops (3) Wrap and tie bandage to maintain pressure. Seek medical care immediately.	--For temporary external use to control traumatic bleeding	Package-- Combat Gauze- [2023]

Supplementary Table 8. Applications of hemostatic materials in porcine liver injury model

Product Name	Applications of hemostatic materials on the injury	How hemostasis was confirmed
FgC6 patch (this study)	Directly applied on the wound with manual compression for 30 s.	The compression was interrupted after 0.5, 1 min to check for the hemostasis.
TachoSil® Fibrin sealant patch	Applied on the wound with manual compression through a moist pad for at least 3 min.	The compression was interrupted after 3, 4, 5, 8 min to check for the hemostasis.
Surgicel® Fibrillar	Hold firmly on the wound for at least 3 min.	The compression was interrupted after 3, 4, 5, 8 min to check for the hemostasis.
Surgiflo®(Thrombin)	Cover the entire bleeding surface and apply a moist pad over the Surgiflo® for at least 2 min.	After 2, 3, 4, 5, 8 min, lift the moist pad to check for the hemostasis.

Supplementary Table 9. Applications of hemostatic materials in porcine femoral artery injury model

Product Name	Applications of hemostatic materials on the injury	How hemostasis was confirmed
FgC6 patch (this study)	Applied on the wound with through a moist pad for 1 min.	The compression was interrupted after 1, 2 min to check for the hemostasis. The survival of injured pigs was observed within 3 h.
Combat Gauze®	Applied on the wound for at least 3 min.	The compression was interrupted after 3, 4, 5, 8 min to check for the hemostasis. The survival of injured pigs was observed within 3 h.

3) The models were also done without following the instructions of the packaging of the hemostatic controls. They did not compress the materials onto the tissues for the required time, for example. In Figures 6 and 13, the competitor dressings were not applied correctly—Combat Gauze was only pressed down for 30 seconds, and Everast for just 10 seconds. While I understand the authors' goal of highlighting their product's ability to stop bleeding within 30 seconds and its lack of long-term effects, the data does not reflect a fair comparison when the competitors' products are not used as intended. Hemostatic treatment typically involves more than just the initial 30 seconds of application.

Response:

Thanks for your suggestions. We have conducted a reassessment of the commercial products in strict adherence to the manufacturer's instructions, ensuring a direct and fair comparison with FgC6 patch in oozing wounds (porcine liver resection injury, 50 mm in length) and massive acute hemorrhage (femoral artery bleeding injury, 6 mm incision). According to instructions of commercial products, Surgicel® Fibrillar, Surgiflo®(Thrombin), EVARREST® and TachoSil® cannot be applied for treatment of major arterial or venous bleeding, and Combat Gauze® is encouraged for massive bleeding. In the revised manuscript, Surgicel® Fibrillar, Surgiflo®(Thrombin), and TachoSil® were used as referential groups in porcine liver resection injury model, and Combat Gauze® was used as a referential group in femoral artery bleeding injury model. For example, TachoSil® was pressed down for at least 3 min in porcine liver resection injury model, and Combat Gauze® was pressed on the wound for at least 3 min in porcine femoral artery injury model. The details of their hemostatic applications have been added in revised Materials and Methods section (Pages 25-27) and Supplementary Tables 7-9.

“Materials and methods

***In vivo* hemostatic performance on porcine liver injury model**

The hemostatic performance was evaluated using porcine liver injury model. This study was approved by the Laboratory Animal Welfare and Ethics Committee of The

Second Affiliated Hospital of Zhejiang University School of Medicine (SAHZU2024#248). Female Bama pigs (22-26 kg, Shanghai Jiagan Biotechnology Co., Ltd) were utilized for porcine liver injury model. General anesthesia was administered for all animal procedures, with maintenance of anesthesia using propofol (0.1-0.2mg/kg/min). The weighed medical gauze was placed beneath the liver. A part of the liver lobe (ca. 50 mm in length) was cut off to establish the porcine liver injury model (two independent injuries per pig; ten pigs). And free bleeding was allowed for 30 s (pretreatment blood loss) to assess bleeding rates. Then the blood was wiped medical gauze from the wound surface before the hemostatic materials were used on the wound. FgC6 patch was directly applied on the wound with manual compression for 30 s. Commercial products (Surgicel[®] Fibrillar, Surgiflo[®](Thrombin), TachoSil[®]) were used according to manufacturer's guidelines (Supplementary Table 7). The compression was interrupted after the certain time intervals to check for the hemostasis (Supplementary Table 8). Each groups contained five independent injuries. When blood did not spread on the medical gauze or around the wound, it was indicative of successful hemostasis. The time was recorded immediately. These hemostatic materials were collected, and absorbed blood on weighed medical gauze or hemostatic materials was recorded as posttreatment blood loss. The pigs were monitored for 3 h, and finally euthanized with a lethal dose of potassium chloride solution under general anesthesia.

***In vivo* hemostatic performance on porcine femoral artery injury model**

The hemostatic performance was evaluated using porcine femoral artery injury model. All animals were treated according to the standard guidelines approved by the Laboratory Animal Welfare and Ethics Committee of The Second Affiliated Hospital of Zhejiang University School of Medicine (SAHZU2024#248). Female Bama pigs (22-26 kg, Shanghai Jiagan Biotechnology Co., Ltd) were utilized for porcine femoral artery injury model. General anesthesia was administered for all animal procedures, with maintenance of anesthesia using propofol (0.1-0.2mg/kg/min). A severe arterial hemorrhage was produced with 6 mm arteriotomy (one injury per pig; twelve pigs), and unrestricted bleeding was allowed for 45 s (pretreatment blood loss) to assess bleeding rates. Then the blood was wiped weighed medical gauze from the inguinal cavity before

the hemostatic materials were used on the wound. FgC6 patch was directly applied on the wound with manual compression for 1 min. Commercial products (Combat Gauze[®]) was used according to manufacturer's guidelines (Supplementary Table 7). The compression was interrupted after the certain time intervals to check for the hemostasis (Supplementary Table 9). Each group contain six independent injuries (pigs). When blood did not spread on the medical gauze or around the wound, it was indicative of successful hemostasis. The time was recorded immediately. These hemostatic materials were collected, and absorbed blood on weighed medical gauze or hemostatic materials was recorded as posttreatment blood loss. The survival of injured pigs was observed within 3 h. After being observed for 3 h, the pigs were finally euthanized with a lethal dose of potassium chloride solution under general anesthesia."

Supplementary Table 7. Methods of application for commercial products from manufacturer’ s guidelines

Product Name	Dosage and administration	Methods of application	Limitations for use/Contraindications	Reference
TachoSil® Fibrin sealant patch	--TachoSil as an adjunct to hemostasis in cardiovascular and hepatic surgery, when control of bleeding by standard surgical techniques (such as suture, ligature or cautery) is ineffective or impractical. --TachoSil is applied to the surface of cardiac, vascular or hepatic tissue only.	(1) Determine the number of patches to be applied by the size of the bleeding area. (2) Select the appropriate TachoSil patch so that it extends 1 to 2 cm beyond the margins of the wound. (3) cleanse the area to be treated. (4) disinfectants and other fluids hold in place with gentle pressure applied through moistened gloves or a moist pad for at least 3 min. (5) To avoid pulling the patch loose, first place a clean surgical instrument at one end of the patch before relieving the pressure	--Not for use in place of sutures or other forms of mechanical ligation in treatment of major arterial or venous bleeding. --Do not apply TachoSil intravascularly. Intravascular application of TachoSil may result in life-threatening thromboembolic events.	Package Insert--- TachoSil [2024]
Surgicel® Fibrillar	--Surgicel Fibrillar is used adjunctively in surgical procedures to assist in the control of capillary, venous and small arterial hemorrhage when ligation or other conventional methods of control are impractical or ineffective.	Use only as much Surgicel Fibrillar as is necessary for hemostasis, holding it firmly in place until bleeding stops.	--Surgicel Fibrillar should not be used to control hemorrhage from large arteries.	Package-Surgicel-[2021]
Surgiflo® (Thrombin)	--Surgiflo, mixed with thrombin solution, is indicated in surgical procedures (other than ophthalmic) as an adjunct to hemostasis when control of bleeding by ligature or other conventional methods is ineffective or impractical.	(1) Identify the source of bleeding. (2) Surgiflo can be used with or without the applicator tip attached to the syringe. Apply sufficient Surgiflo Hemostatic Matrix to cover the entire bleeding surface.	--Do not use Surgiflo in intravascular compartments because of the risk of embolization. --Do not use Surgiflo in closure of skin incisions because it may interfere with the healing of skin edges.	Package-Surgiflo [2020]

		(3) Apply a sterile saline moistened gauze over the Surgiflo to ensure the material remains in contact with the bleeding tissue. (4) After 1-2 min, lift the gauze and inspect the wound site. Once bleeding has ceased, irrigate excess Surgiflo away gently so as not to disturb the new clot.	--Surgiflo should not be used in instances of pumping arterial hemorrhage. It should not be used where blood or other fluids have pooled or in cases where the point of hemorrhage is submerged.	
Combat Gauze®	--Penetrating wounds and massive bleeding --Gunshot wounds --Stabbings --Large non-compressible injuries where tourniquets cannot be applied	(1) Pack Combat Gauze into wound and use it to apply pressure directly over bleeding source (2) Continue to apply pressure for 3 min or until bleeding stops (3) Wrap and tie bandage to maintain pressure. Seek medical care immediately.	--For temporary external use to control traumatic bleeding	Package-- Combat Gauze- [2023]

Supplementary Table 8. Applications of hemostatic materials in porcine liver injury model

Product Name	Applications of hemostatic materials on the injury	How hemostasis was confirmed
FgC6 patch (this study)	Directly applied on the wound with manual compression for 30 s.	The compression was interrupted after 0.5, 1 min to check for the hemostasis.
TachoSil® Fibrin sealant patch	Applied on the wound with manual compression through a moist pad for at least 3 min.	The compression was interrupted after 3, 4, 5, 8 min to check for the hemostasis.
Surgicel® Fibrillar	Hold firmly on the wound for at least 3 min.	The compression was interrupted after 3, 4, 5, 8 min to check for the hemostasis.
Surgiflo®(Thrombin)	Cover the entire bleeding surface and apply a moist pad over the Surgiflo® for at least 2 min.	After 2, 3, 4, 5, 8 min, lift the moist pad to check for the hemostasis.

Supplementary Table 9. Applications of hemostatic materials in porcine femoral artery injury model

Product Name	Applications of hemostatic materials on the injury	How hemostasis was confirmed
FgC6 patch (this study)	Applied on the wound with through a moist pad for 1 min.	The compression was interrupted after 1, 2 min to check for the hemostasis. The survival of injured pigs was observed within 3 h.
Combat Gauze®	Applied on the wound for at least 3 min.	The compression was interrupted after 3, 4, 5, 8 min to check for the hemostasis. The survival of injured pigs was observed within 3 h.

4) The manuscripts noted that Combat Gauze is not considered a hospital standard, however it is used extensively for abdominal packing and other surgical applications.

Response:

Thanks for pointing out this question. The statement about Combat Gauze[®] has been corrected in the revised manuscript.

“Combat Gauze[®], which is recommended for massive hemorrhage in Tactical Combat Casualty Care (TCCC) Guidelines⁴⁶, was used as a reference group (Supplementary Table 6). Combat Gauze[®] need be applied with at least 3 min of direct pressure to control bleeding.”

Reviewer #2:

The authors have responded to my previous comments and I have no further questions.

Response: We appreciate the reviewer's comments of our work. The comments are all valuable and helpful for improving our article.

We would like to thank the reviewers again for taking the time to review our manuscript.

REVIEWERS' COMMENTS

Reviewer #1 (Remarks to the Author):

My concerns have been sufficiently addressed.

Response: We sincerely thank the referee for recommending the publication of our paper. The previous constructive comments are valuable and helpful for improving our article. We hope this work could contribute to developing a protein patch for biocompatible and biodegradable haemostatic sealing, alleviating patient discomfort and enhancing surgical efficiency.